# Subpixel object segmentation using wavelets and multi resolution analysis

## Abstract

We propose a novel deep learning framework for fast prediction of boundaries of two-dimensional simply connected domains using wavelets and Multi Resolution Analysis (MRA). The boundaries are modelled as (piecewise) smooth closed curves using wavelets and the so-called Pyramid Algorithm. Our network architecture is a hybrid analog of the U-Net, where the down-sampling path is a two-dimensional encoder with learnable filters, and the upsampling path is a one-dimensional decoder, which builds curves up from low to high resolution levels. Any wavelet basis induced by a MRA can be used. This flexibility allows for incorporation of priors on the smoothness of curves. The effectiveness of the proposed method is demonstrated by delineating boundaries of simply connected domains (organs) in medical images using Debauches wavelets and comparing performance with a U-Net baseline. Our model demonstrates up to 5x faster inference speed compared to the U-Net, while maintaining similar performance in terms of Dice score and Hausdorff distance.

## 1 Introduction

Semantic image segmentation is a core component of many medical imaging related tasks. Both as part of a pipeline to find a region of interest, or a task by itself, e.g., for measuring tumor volume. Nowadays, almost all segmentation algorithms in medical imaging are replaced by U-Net-like architectures Ronneberger et al. (2015) combining an encoder and decoder. Typically, the decoder is an upsampling path, and additional skip connections between the encoding and decoding part are added to recover the image's spatial information. While many variants or more exotics methods, such as multi-scale and pyramid based approaches, recurrent networks or generative techniques, can be designed, all of these still yield per-pixel classifications. In these settings an image is interpreted as a *discrete* collection of ordered pixels (or voxels in the three-dimensional case), where the task is to assign an appropriate class to each pixel using a probabilistic model. Typically, the pixels are assumed to be independent so that the joint-likelihood is tractable.

In practice, e.g., in medical imaging, the boundary of a region is annotated and not the region itself. Hence the raw ground-truth data is a discretization of a closed curve. The main motivation of our paper is to construct a deep learning model which $(i)$ may directly use such raw ground-truth data if available $(ii)$ is guaranteed to predict smooth planar curves $(iii)$ improves inference speed by predicting 1d objects (curves) instead of 2d objects. We argue that in traditional pipelines, where pixel-based predictions are constructed, smooth boundaries are not faithfully represented. In particular, no prior information about the geometry of planar curves is incorporated. In this paper, we present a hybrid analog of the U-Net, where the down-sampling path is a two-dimensional encoder with learnable filters, and the upsampling path is an one-dimensional decoder, which predicts (smooth) representations of curves.

A fundamental component in the setup of our framework is the decision on how to represent (closed) curves. While the Fourier basis is a natural candidate at first glance, its global nature may hamper accurate predictions of curves which exhibit highly localized behavior, requiring accurate estimates of small noisy high-frequency modes. For this reason, we have chosen to represent contours using wavelets and Multi Resolution Analysis (MRA) instead. The main idea is to choose a single map $\varphi$, the so-called scaling function or father wavelet, and to construct subspaces of functions associated to prescribed resolution levels by taking the span of appropriate dilations and translations of $\varphi$. This

setup provides an efficient way to decompose and reconstruct contours, from low to high resolution level, using the classical Pyramid Algorithm Mallat (2008) as a decoder. The filters in the decoder are not learned but uniquely determined by the chosen wavelet basis. Any wavelet basis induced by a MRA can be used. This flexibility allows for incorporation of priors on the smoothness of curves.

**Related work**  Previous work by Chen et al. (2019); Marcos et al. (2018); Hatamizadeh et al. (2020) also proposed models to predict contours by combining Active Contour Models (ACM) with a CNN into an end-to-end model. In these papers the representation of curves is ultimately still based on pixel-based computations. For instance, in Hatamizadeh et al. (2020) curves are modelled as level sets of distance maps defined on a discretization of the domain of the image. In Chen et al. (2019) a similar approach is followed, but a smoothed approximation of an indicator function is used instead of a distance map. The work in Marcos et al. (2018) is perhaps most closely related to ours; they directly construct polygonal approximations of curves and represent them using (pixel) coordinates of the nodes. The objectives minimized in the cited papers are based on a careful consideration of mean pixel intensities and geometric properties such as area and arc length. These properties are implicitly encoded in an objective function (energy-functional) defined on a space of distance maps Hatamizadeh et al. (2020), suitable approximations of indicator functions Chen et al. (2019), or family of polygons Marcos et al. (2018). However, in contrast to our approach, the above methods all provide pixel-based output. To the best of our knowledge, our work is the first to use MRA and wavelet analysis to construct pixel-independent representations of (closed) curves.

This paper is organized as follows. In Section 2 we review the mathematical background needed to construct our model. We explain how contours can be decomposed and reconstructed on different resolution levels. The reconstruction algorithm, the Pyramid Algorithm, forms a core component of our network architecture. In Section 3 we set up the model architecture and loss. Subsequently the datasets, training method and performance measures are described in Section 4. We end the paper with results and a discussion in Section 5. Further mathematical details are provided in the appendix.

## 2 BACKGROUND AND MATHEMATICAL SETUP

In this section we describe our mathematical setup and review the theory needed to construct our model. We consider two dimensional gray-valued images $x \in \mathcal{X} := [0,1]^{n \times n}$, e.g., slices of MRI scans of size $n \times n$, where $n \in \mathbb{N}$. We assume that each image contains a (uniquely identifiable) simply connected region $R(x) \subset \mathbb{R}^2$, e.g., an organ, with boundary $\partial R(x)$. It is assumed that $\partial R(x)$ can be parameterized by a simple closed continuous curve $\gamma(x)$. We will develop a deep learning framework for computing such parameterizations $\gamma(x)$ using Multi Resolution Analysis (MRA). Since wavelets play a seminal role in our set up, we first review the necessary theory in Sections 2.1 and 2.2 as a subject in its own right in the context of general scalar-valued signals $\gamma$. In the remainder of the paper, we return to the context in which $\gamma = \gamma(x)$ is interpreted as a periodic (planar) curve parameterizing the boundary of a region.

### 2.1 MULTI RESOLUTION ANALYSIS

In this section we briefly review wavelet theory using the framework of a multi resolution analysis (MRA). We closely follow the exposition in Pereyra & Ward (2012) and Montefusco & Puccio (2014). Throughout this section we denote the space of square integrable functions on $\mathbb{R}$, equipped with standard inner product, by $L^2(\mathbb{R})$.

The uncertainty principle in Fourier analysis dictates that a signal $\gamma \in L^2(\mathbb{R})$ cannot be simultaneously localized in the time and frequency domain, see Pereyra & Ward (2012). Multi resolution analysis aims to address this shortcoming by decomposing a signal on different *discrete* resolution levels. The idea is to construct subspaces $V_j \subset L^2(\mathbb{R})$, associated to various resolution levels $j \in \mathbb{Z}$, spanned by integer shifts of a localized mapping $\varphi_j$. The level of localization associated to $V_j$ is determined by taking an appropriate dilation of a prescribed map $\varphi$; the so-called *scaling function*. In the MRA framework the dilation factors are chosen to be powers of two. Formally, we require that $(\varphi_{jk})_{k \in \mathbb{Z}}$ is an orthonormal basis for $V_j$, where $\varphi_{jk}(t) := 2^{\frac{j}{2}} \phi(2^j t - k)$, see Figures 1a and

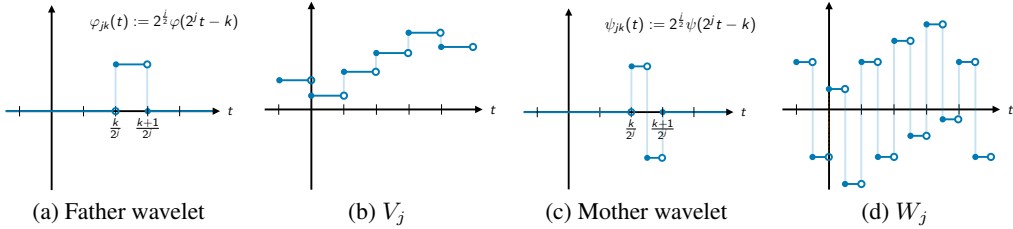

|  |  |  |  |
|---|---|---|---|
| (a) Father wavelet | (b) $V_j$ | (c) Mother wavelet | (d) $W_j$ |

Figure 1: Example of the Haar MRA: (a) Dilated translation of the Haar scaling map $\varphi = \mathbf{1}_{[0,1)}$. (b) The approximation subspace at level $j$ consists of all step-functions with step-size $2^{-j}$. (c) Dilated translation of the mother wavelet $\psi = \mathbf{1}_{[0,\frac{1}{2})} - \mathbf{1}_{[\frac{1}{2},1)}$. (d) Example of a function in the detail subspace at level $j$.

.

1b. Altogether, this yields an increasing sequence of closed subspaces $V_j \subset V_{j+1} \subset L^2(\mathbb{R})$ dense in $L^2(\mathbb{R})$, where $V_{j+1}$ is the next level up in resolution after $V_j$.

The representation of $\gamma$ at resolution level $j$, denoted by $\gamma_j := P_j\gamma$, is its orthogonal projection onto $V_j$. Here $P_j$ denotes the orthogonal projection onto $V_j$. The coefficients of $\gamma_j$ with respect to the basis for $V_j$, denoted by $a_j(\gamma) = (a_{jk}(\gamma))_{k\in\mathbb{Z}}$, are called the *approximation coefficients*. The associated subspaces $V_j$ are referred to as the approximation subspaces (see Figure 1b). To study the information that is lost when a signal in $V_{j+1}$ is projected onto $V_j$, we consider the operator $Q_j := P_{j+1} - P_j$. The range of $Q_j$, denoted by $W_j$, is referred to as the detail subspace at level $j$; it is the orthogonal complement of $V_j$ in $V_{j+1}$. The detail subspaces $(W_j)_{j\in\mathbb{Z}}$ are mutually disjoint and orthogonal by construction. A fundamental result, known as Mallat's Theorem, states that the subspaces $W_j$ can too be spanned by dilating and shifting a single map. More precisely, there exists a map $\psi \in W_0$, the so-called *mother wavelet*, such that $(\psi_{jk})_{k\in\mathbb{Z}}$ is an orthonormal basis for $W_j$, see Pereyra & Ward (2012). The coefficients of $Q_j\gamma$ with respect to this basis, denoted by $d_j(\gamma) := (d_{jk}(\gamma))_{k\in\mathbb{Z}}$, are referred to as the *detail coefficients* of $\gamma$ at resolution level $j$. The detail coefficients store the information needed to go back one level up in resolution, since $P_{j+1} = P_j + Q_j$ by construction. We often write $a_j(\gamma) = a_j$ and $d_j(\gamma) = d_j$ for brevity. In practice, we only approximate a finite number of approximation and detail coefficients, see Section 2.3.

## 2.2 THE DISCRETE WAVELET TRANSFORM

In this section we describe how to compute the approximation and detail coefficients given a pre-scribed scaling function $\varphi$. Many fundamental aspects of MRA's, both theoretical and compu-tational, can be traced back to the following key observation. Since $V_0 \subset V_1$, there must exist coefficients $h = (h_k)_{k\in\mathbb{Z}}$ such that $\varphi = \sum_{k\in\mathbb{Z}} h_k\varphi_{1k}$. This equation is referred to as the *scaling equation*; one of the fundamental properties of a scaling function. The sequence $h$, the so-called low-pass filter, completely characterizes the scaling function. Similarly, since $\psi \in W_0 \subset V_1$, there exist coefficients $g = (g_k)_{k\in\mathbb{Z}}$, the so-called high-pass filter associated to $h$, such that $\psi = \sum_{k\in\mathbb{Z}} g_k\varphi_{1k}$. For Mallat's mother wavelet, we have $g_k = (-1)^{k-1}\overline{h_{1-k}}$. Altogether, in order to define a MRA, one only needs to specify an appropriate low-pass filter $h$.

The scaling equations can be used to derive an efficient scheme for computing lower order approx-imation coefficients (of any order) given an initial approximation $a_{j+1}$. Conversely, the orthogonal decomposition $V_{j+1} = V_j \oplus W_j$ can be used to reconstruct $a_{j+1}$ given the approximation and de-tail coefficients $a_j$ and $d_j$, respectively, at resolution level $j$. The computations are summarized in Figure 2. The reconstruction and decomposition formulae together form the well-known Pyramid Algorithm Mallat (2008).

In practice, our signals are periodic and do not directly fit into the MRA framework, since non-zero periodic signals are not elements in $L^2(\mathbb{R})$. We will address this issue in the next section by using an appropriate cut-off. Here we only remark that the periodicity needs to be carefully taken into account in the decomposition and reconstruction formulae, see Appendix A.2.

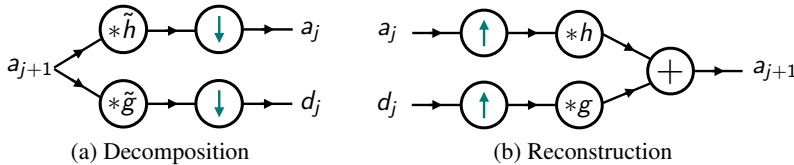

(a) Decomposition          (b) Reconstruction

Figure 2: (a) Decomposing approximation coefficients at level $j + 1$ into approximation and detail coefficients at level $j$. Here $\tilde{h}$ and $\tilde{g}$ are defined by $\tilde{h}_k := \overline{h_{-k}}$ and $\tilde{g}_k := \overline{g_{-k}}$, respectively, $*$ is the two-sided discrete convolution and $\downarrow$ downsamples a sequence by discarding all terms with odd index. (b) Reconstructing approximation coefficients at level $j + 1$ from the approximation and detail coefficients at level $j$. Here $\uparrow$ samples a sequence up by putting zeros in between every term.

## 2.3 WAVELET REPRESENTATION OF PERIODIC CURVES

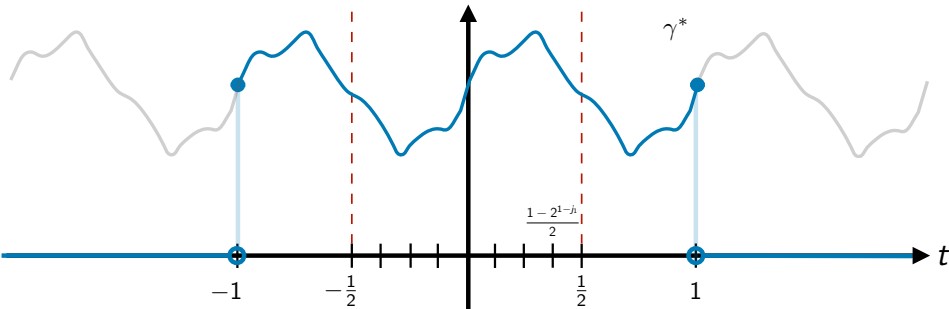

Figure 3: The cut-off signal $\gamma^*(t) = \gamma(lt)\mathbf{1}_{[-1,1]}(t)$ depicted in blue. We only compute approximation coefficients associated to the region $[-\frac{1}{2}, \frac{1}{2}]$.

In this section we explain how a scalar-valued periodic signal $\gamma$ with period $l > 0$ can be approximated using a MRA. Let $j_1 \in \mathbb{N}$ be a desired resolution-level. First, we re-parameterize $\gamma$ to have period 1. The reason for this is rather technical and we refer the reader to Appendix A.1. In essence, we require the number of approximation coefficients to be a power of two and this parameterization fits nicely with this requirement. Next, to address the issue that periodic signals are not contained in $L^2(\mathbb{R})$, we restrict the re-parameterized curve to $[-1, 1]$, i.e., set $\gamma^*(t) := \gamma(lt)\mathbf{1}_{[-1,1]}(t)$. In general, this will introduce discontinuities at the boundary points $-1$ and $1$. This is, however, not an issue, since we only need information about $\gamma^*$ on a strict subset $[I_0, I_1] \subset [-1, 1]$ of length 1.

It is shown in Lemma A.1 how (and which) approximation coefficients can be related to the sample values of $\gamma$. In particular, if $j_1$ is sufficiently large, the approximation coefficients needed to (approximately) cover $[-1, 1]$ are $(a_{j_1 k}(\gamma^*))_{k=-2^{j_1}}^{\lfloor 2^{j_1} - \beta \rfloor}$. Here $\beta > 0$ is the support of the underlying wavelet. These coefficients will be close to the (scaled) sample values of $\gamma$ on $\{k2^{-j_1} : -2^{j_1} \leq k \leq \lfloor 2^{j_1} - \beta \rfloor\}$. Motivated by this observation, and the fact that we only need $\gamma$ on $[-\frac{1}{2}, \frac{1}{2}]$, we use the coefficients $(a_{j_1 k}(\gamma^*))_{k=-2^{j_1-1}}^{2^{j_1-1}-1}$ only, which cover $[-\frac{1}{2}, \frac{1-2^{1-j_1}}{2}]$ approximately, see Figure 3. To ensure that $2^{j_1-1} - 1 < \lfloor 2^{j_1} - \beta \rfloor$, we require that $j_1 \geq \left\lceil \frac{\log(\beta-1)}{\log(2)} + 1 \right\rceil$. This quantity is well-defined for all bases considered in this paper, except the Haar-basis.

## 3 MODEL

In this section we formulate our objective and network architecture. Henceforth $\gamma = \gamma(x)$ is associated to an image $x$ and has two components $[\gamma(x)]_1$ and $[\gamma(x)]_2$. All operations from the previous sections are understood to be carried out component-wise.

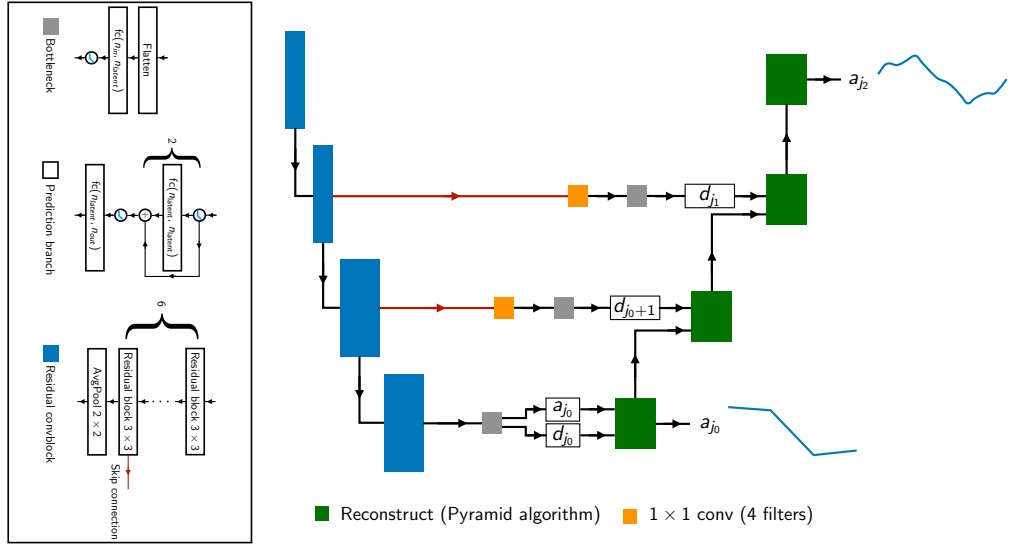

Figure 4: A schematic picture of our network. Here a "residual block" corresponds to a standard residual convolutional layer. The first residual conv-block uses 32 filters and is doubled after every two residual conv-blocks. Each green block corresponds to the operation depicted in Figure 2b. In the decoder, we only predict detail coefficients up to level $j_1$. No detail coefficients are used at levels $j_1 + 1 \leq j \leq j_2$. In this example, we have set $j_1 = j_0 + 2$ and $j_2 = j_0 + 3$. In reality, the decoder consists of two upsampling paths, one for each spatial component of the curve. We have only drawn one for notational convenience. During training, only the approximation coefficients at the lowest and highest resolution levels (most right curves in blue) are supervised.

## 3.1 NETWORK ARCHITECTURE

To formulate our objective, consider an image $x \in \mathcal{X}$ with an associated boundary $\partial R(x)$ of a simply connected region $R(x)$. We assume that $\partial R(x)$ is parameterized by a closed continuous curve $\gamma(x)$ of length $l(x) > 0$. The objective is to compute the *relevant* approximation coefficients of $\gamma^*(x)$, see the explanation in Section 2.3, where $\gamma^*(x)$ denotes the cut-off re-parameterization of $\gamma(x)$, using a suitable neural network.

Our network is a hybrid analog of the U-net. It consists of an encoder, bottleneck and decoder with skip-connections in between. Only the approximation coefficients at the lowest resolution-level $j_0$ are "directly" computed by the network (in the bottleneck). Afterwards, the Pyramid Algorithm takes over to compute approximation coefficients on higher resolution levels (the decoder). In practice, the detail coefficients are negligible on sufficiently high resolution levels. For this reason, we only predict detail coefficients up to a prescribed level $j_1$. The predictions at higher resolution levels $j_1 < j \leq j_2$ are computed without detail coefficients. The full architecture is summarized in Figure 4.

## 3.2 LOSS

The loss $\mathcal{L}$ consists of two parts: ordinary cross-entropy for optimizing the likelihood of $p(x)$ and a part corresponding to the $L^2$-error between observed and predicted curves on different resolution levels. More precisely, set $r(x) = 1$ if $R(x) \neq \emptyset$ and $r(x) = 0$ otherwise, then $\mathcal{L}_{ce}(p(x), r(x)) := r(x) \log p(x) + (1 - r(x)) \log(1 - p(x))$. Next, suppose $a([\gamma^*(x)]) = (a_{j_0}([\gamma^*(x)]), \ldots, a_{j_2}([\gamma^*(x)]))$ are the approximation coefficients of $\gamma^*(x)$ on resolution levels $j_0 \leq j \leq j_2$. Define $\mathcal{L}_{js}(f_j(x), a_j(\gamma^*(x))) := \|[f_j(x)]_s - a_j([\gamma^*(x)]_s)\|_2$ for $s \in \{1, 2\}$ and $j \in \{j_0, j_2\}$. Here $f_j(x)$ are the network predictions of the relevant approximation coefficients at level $j$. This loss-term corresponds to the $L^2$-error on resolution level $j$ between the curves with

approximation coefficients $[f_j(x)]_s$ and $a_j([\gamma^*(x)]_s)$. Finally, define the total loss by

$$\mathcal{L}(F(x), a) := w\mathcal{L}_{\text{ce}}(p(x), r(x)) + r(x) \sum_{\substack{s \in \{1,2\} \\ j \in \{j_0, j_2\}}} \mathcal{L}_{js}(f_j(x), a_j(\gamma^*(x))),$$

where $w > 0$ is a weight. Notice that $\mathcal{L}$ measures the discrepancies between observed and predicted curves on the lowest and highest resolution levels only. In practice, this enforces the approximation and detail coefficients at intermediate levels to agree as well; see the experiments in Section 5.

## 4 TRAINING

In this section we describe the dataset on which we test our method. In addition, we provide details about preprocessing steps and model development (training).

### 4.1 DATASETS

**Toy dataset**   The main purpose of the toy-example is to create a setting in which the annotated contours differ substantially from annotations confined to a grid. For this purpose, we consider piecewise smooth curves having a finite number of non-differentiable points. The toy-dataset consists of hypocycloids, up to an Euclidian motion and scaling, defined by

$$\eta(t) := \left[ (r_1 - r_2)\cos t + r_2 \cos\left(\frac{r_1 - r_2}{r_2}t\right) \quad (r_1 - r_2)\sin t - r_2 \sin\left(\frac{r_1 - r_2}{r_2}t\right) \right]^T,$$

where $r_1 > r_2$ and $t \in \mathbb{R}$. If $\frac{r_1}{r_2} \in \mathbb{N}$, then $\eta$ is closed and has exactly $\frac{r_1}{r_2}$ cusps (non-differentiable points). To easily control the number of cusps, we fix $r_2 = 1$ and vary $r_1$. Note that in this case $\eta$ has $r_1$ cusps and period $2\pi$.

We construct curves and binary masks of various sizes, orientation and positions, by sampling a radius $r_1 \in \{3, 4, 5, 6\}$ from $\mathcal{U}(\{3, 4, 5, 6\})$, angle $\theta$ from $\mathcal{U}([-\frac{\pi}{2}, \frac{\pi}{2}])$, components $q_1, q_2$ from $\mathcal{U}([-80, 80])$ for a shift, and a scaling factor $\kappa$ from $\mathcal{U}([10, 20])$. Here $\mathcal{U}(I)$ denotes a random variable uniformly distributed on $I$, where $I$ is an interval of finite length or a discrete finite set. Next, we evaluate the curve $\kappa\left(R(\theta)\eta + [160 + q_1 \quad 160 + q_2]^T\right)$ on an equispaced grid of $[0, 2\pi]$ of size 512. Here $R(\theta)$ corresponds to an anti-clockwise rotation around the origin with angle $\theta$. Finally, the discretized curve is used to construct a binary mask of size $320 \times 320$ using SKIMAGE.

**Medical decathlon**   The data used to evaluate the performance of our model consists of MRI images of the prostate central gland, henceforth abbreviated as just the prostate, and CT scans of the spleen. The datasets are part of a public dataset made available for the Medical Decathlon Contest Simpson et al. (2019) . The dataset for the prostate consists of T2-weighted MRI images of size $320 \times 320$, which were cropped to size $224 \times 224$. The dataset for the spleen consists of CT scans of size $512 \times 512$ and was cropped to size $256 \times 256$. The cropping was based on constructing bounding boxes of the form $[u_{\min} - \delta_p, u_{\max} + \delta_p] \times [v_{\min} - \delta_p, v_{\max} + \delta_p]$ for the training set, where $u_{\min}, v_{\min}, u_{\max}$ and $v_{\max}$ are the minimal and maximal coordinates of the segmentation in each direction, respectively, using an offset of $\delta_p = 65$ pixels. A residual CNN (encoder of five blocks) was trained (and validated) on the training set to regress the corner and center points of the bounding boxes using a RMSE-loss. This rather crude approach is not meant to produce tight bounding boxes, but serves as a rough necessary localization step to improve performance, and allows us to focus on the task of shape-prediction only.

### 4.2 CONSTRUCTION GROUND-TRUTH

In this section we describe how the ground-truth data is generated using the Pyramid Algorithm and Lemma A.1. Let $(x, u) \in \mathcal{X} \times \mathbb{R}^{n_s \times n_p}$ be an image (slice) - contour pair, where $x$ is a slice of the CT or MRI scan, $u$ is a finite sequence of points approximating a closed curve and $n_s = 2$ is the number of spatial components. Since we only have access to binary masks (for the public datasets), and not to the raw annotations themselves, we extract $u$ using OPENCV. While not ideal, we stress that $u$ contains "subpixel" information and is *not* constrained to an integer-valued grid.

**Fourier coefficients**    To initialize the Pyramid Algorithm, we compute the approximation coefficients at level $j_1$ using Lemma A.1. To accomplish this, we need to compute a Fourier expansion for $u$. First, we parameterize the contour by arc length. The arc length $l$ is approximated by summing up the Euclidian distances between subsequent points in $u$. The Fourier coefficients are then computed by evaluating the contour on an equispaced grid of $[0, l]$ of size $2N - 1$, where $N \in \mathbb{N}$, using linear interpolation and the Discrete Fourier Transform. Since the contours are real-valued, we only store the Fourier coefficients $(\tilde{\gamma}_m)_{m=0}^{N-1} \in (\mathbb{C}^{n_s})^N$. Fourier coefficients that are too small, i.e., have no relevant contribution, are set to zero; see Appendix B.1 for the details.

**Consistency**    To have consistent parameterizations for all slices, we ensure that $u$ is always traversed anti clock-wise (using `opencv`). Furthermore, since the parameterization is only determined up to a translation in time, we need to pick out a specific one. We choose the unique parameterization such that the contour starts at angle zero at time zero relative to the midpoint $c = (c_1, c_2) \in \mathbb{R}^2$ of $R$. The implementation details are provided in Appendix B.2.

**Approximation and detail coefficients**    Altogether, the above steps yield a contour $\gamma$ with Fourier coefficients $(\gamma_m)_{m=1-N}^{N-1}$ and period $l$. We reparameterize $\gamma$ to have period 1, as explained in Section 2.3, and "center" the contour using the average midpoint computed over the training-set. The initial approximation coefficients at level $j_2$ are then computed using Lemma A.1. Next, the Pyramid Algorithm is used to compute approximation and detail coefficients at levels $j_0 \leq j \leq j_2 - 1$. We set the detail coefficients which are in absolute value below $\varepsilon = 5 \cdot 10^{-3}$ to zero to reduce noise. Subsequently, we reconstruct the approximation coefficients at levels $j_0 + 1 \leq j \leq j_1$ using the thresholded detail coefficients. No detail coefficients are used to compute approximation coefficients at levels $j_1 + 1 \leq j \leq j_2$. The final approximation and (thresholded) detail coefficients $a$ and $d$, respectively, are used as ground-truth.

The resulting dataset $\mathcal{D}$ thus consists of tuples $(x, a, d)$. The training and validation set $\mathcal{D}_{\text{train}}$ and $\mathcal{D}_{\text{val}}$, respectively, are obtained by randomly omitting subjects from the the full dataset $\mathcal{D}$. For the toy-example, spleen and prostate we have $(|\mathcal{D}_{\text{train}}|, |\mathcal{D}_{\text{val}}|) = (1650, 250)$, $(|\mathcal{D}_{\text{train}}|, |\mathcal{D}_{\text{val}}|) = (527, 75)$, and $(|\mathcal{D}_{\text{train}}|, |\mathcal{D}_{\text{val}}|) = (3148, 502)$, respectively. Before feeding the images $x$ into the model, we linearly rescale the image intensities at each instance to $[0, 1]$. Furthermore, for the spleen and prostate, we use extensive data augmentation: we use random shifts, random rotations, random scaling, elastic deformations, horizontal shearing and random cropping.

### 4.3 Model training

We use the Adam optimizer Kingma & Ba (2015) to train our network for 150 epochs. In the first five epochs, we use a linearly increasing learning rate from $5 \cdot 10^{-4}$ to $10^{-3}$, which is subsequently decayed by a factor of $0.5$ each time the loss does not significantly decrease for 10 subsequent epochs (for the remaining 145 epochs). A batch size of 16 samples is used in each descent step. Finally, the model with the lowest loss is selected. The computations were performed in PYTORCH on a Geforce RTX 2080 Ti.

## 5 Results

In this section we examine the performance of our method and its dependence on the choice of basis. We fix all other hyper-parameters (as much as possible). We consider the Debauches wavelets db$p$ with $p \in \{1, 2, 4, 8, 16\}$ vanishing moments. Roughly speaking, db$p$ corresponds to the unique MRA for which the mother wavelet has minimal support and $p$ vanishing moments. For each basis, we fix appropriate resolution levels $j_0$, $j_1$ and $j_2$ as follows. We fix $j_0$ as the smallest possible resolution level for which the length of our signal ($2^{j_0}$) still exceeds the length of the low-pass filter. Since db$p$ has a low-pass filter of length $2^p$, this forces $j_0(p) = p$. The choice for $j_1 \geq j_0$ is based on the observations that the norm of $d_j$ decreases as $j$ increases and small detail coefficients have no significant impact anymore (see Figure 11). We fix $j_1 \geq j_0$ as the smallest level for which $|[d_{j_1 k}]_s| < 5 \cdot 10^{-3}$ for all samples in the training set (also see the discussion in Section 4.2). The final resolution level $j_2$ and the support of the mother wavelet determine how much of the contour is covered, see Figure 3. We therefore fix $j_2 \geq j_1$ as the smallest resolution level for which the distance between the end-points of the curves are within 1 pixel distance for all samples in the training set.

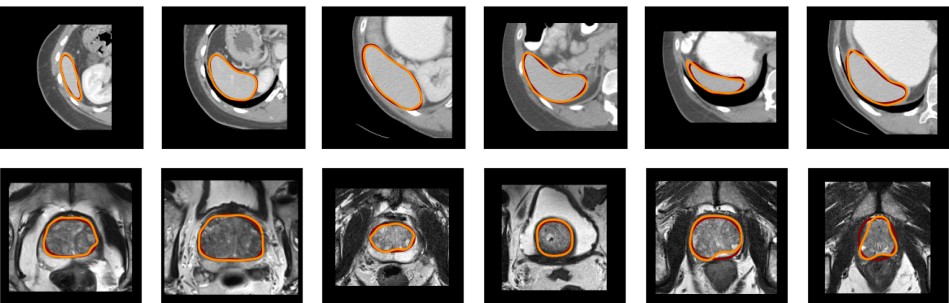

Figure 5: Predicted and observed boundaries colored in red and orange, respectively, for the spleen and prostate for the best models (db8). The last two columns correspond to "hard" examples. Predictions for the other wavelet bases, as well as the toy-example, can be found in Appendix D.

We use $64$ Fourier coefficients for the the toy-problem, spleen and prostate, respectively to initialize the approximation coefficients.

**Baseline**    For the prostate and spleen we use the U-Net in Ronneberger et al. (2015) and a recently developed variant in Nikolov et al. (2018) for comparison. We have modified the network architectures to match the parameter count with our networks (approximately), while maintaining the structure and idea's presented in the original papers as much as possible, see Appendix C. We stress, however, that our objective, i.e., parameterizing contours, is different from the U-Net's objective. The binary ground-truth matched by a U-Net is a fundamentally different (often easier) object than the continuous representation of a curve matched by our networks. Subtle curvature and geometry may be accurately presented using our ground-truth curves, e.g., by using a sufficiently large number of Fourier coefficients to compute approximation coefficients. Binary ground-truth masks, however, cannot capture such subtle geometry due to their discrete nature.

**Performance measures**    We evaluate accuracy using two-dimensional quantities only, since our models are 2d. We compute the component-wise $L^2$-errors between observed and predicted curves on the highest resolution level $j_2$ by taking the $\ell^2$-norm of the approximation coefficients. Furthermore, we compute the dice score and Haussdorf distance between curves using the implementation in SHAPELY. This requires a polygonal approximation of the contour, which is easily obtained using the approximation coefficients at level $j_2$. Note that the Hausdorff distance between subsets of a general metric space may differ from the Hausdorff distance between the associated boundaries (wavelet models), but coincide for (compact) simply connected subsets of $\mathbb{R}^n$. The results are shown in Table 1. The predictions for the best performing wavelet models with respect to the Hausdorff-score are shown in Figure 5. Predictions for other bases, corresponding wavelet decompositions, and detailed visualization of statistics (violin plots) can be found in Appendix D.

**Toy problem**    We observe that all models perform well and are capable of accurately parameterizing piecewise smooth curves. For db1, however, we observe relatively large gaps between the end-points, since its support is relatively small. In addition, db1 has difficulty with accurately predicting small "densely" sampled cycloids. In general, the predictions associated to the less regular wavelets db1 and db2 sometimes exhibit "small" oscillatory behavior. We found that the latter two issues were caused by a too large resolution level $j_2$. To see why, note that the features extracted from the images only contain information up to a certain resolution level. The subpixel information needed to "fill in the blanks", so to speak, is in part provided by the ground-truth data and in part by the chosen wavelet basis. The regularity and support of the wavelet determines to which extent, i.e., up to which resolution level, subpixel information can be "filled in". As the regularity (and support) of the wavelet decreases, the maximal achievable resolution level decreases as well.

**Spleen**    The predictions of our models are accurate and on par with the baseline U-Nets. The more advanced U-Net in Nikolov et al. (2018) performs slightly better; this is mostly due to "edge" cases where the boundary of the spleen is small and about to disappear from our two-dimensional sliced-view. In such cases it may sometimes be ambiguous to define an accurate ground-truth contour,

Table 1: Mean and standard deviation of various performance measures for the toy example, spleen and prostate. The standard deviation is reported in parentheses. The column $N_p$ is the approximate number of model parameters in millions, $T$ is the inference time per image in milliseconds, and db$p$- refers to a network trained without detail coefficients. The length of the encoder for the toy-example is six and five for the prostate and spleen.

| | Model | $(j_0, j_1, j_2)$ | Dice | Hausdorff | $L^2$ $(s=1)$ | $L^2$ $(s=2)$ | $N_p$ | $T$ |
|---|---|---|---|---|---|---|---|---|
| **Toy** | db1 | $(1,6,8)$ | 0.962 (0.028) | 3.520 (1.804) | 0.846 (0.317) | 0.830 (0.281) | 7.70 | 15.6 |
| | db2 | $(2,6,8)$ | 0.971 (0.021) | 2.330 (0.889) | 0.377 (0.172) | 0.373 (0.161) | 5.96 | 15.1 |
| | db4 | $(3,6,8)$ | **0.980** (0.014) | 1.317 (0.509) | 0.264 (0.122) | 0.267 (0.124) | 5.45 | 14.5 |
| | db8 | $(4,6,8)$ | 0.978 (0.016) | **1.287** (0.520) | 0.276 (0.149) | 0.281 (0.162) | 5.25 | 14.1 |
| | db16 | $(5,6,8)$ | 0.978 (0.018) | 1.469 (0.503) | **0.252** (0.104) | **0.254** (0.111) | 5.26 | **14.0** |
| | db2- | $(2,-,8)$ | 0.787 (0.036) | 13.623 (4.868) | 2.968 (0.758) | 3.015 (0.916) | 5.11 | 14.9 |
| | db4- | $(3,-,8)$ | 0.873 (0.165) | 7.222 (6.699) | 1.239 (0.569) | 1.213 (0.541) | 5.11 | 14.3 |
| | db8- | $(4,-,8)$ | 0.967 (0.013) | 2.215 (0.845) | 0.432 (0.170) | 0.430 (0.177) | 5.11 | 14.0 |
| | db16- | $(5,-,8)$ | 0.977 (0.018) | 1.525 (0.516) | 0.264 (0.104) | 0.264 (0.110) | **5.11** | 14.0 |
| **Spleen** | db1 | $(1,5,8)$ | 0.926 (0.034) | 5.203 (2.040) | 2.543 (0.997) | 2.915 (1.276) | 9.92 | 11.6 |
| | db2 | $(2,5,8)$ | 0.943 (0.030) | 4.200 (1.680) | **1.798** (1.170) | 2.113 (1.322) | 5.62 | 10.5 |
| | db4 | $(3,5,8)$ | 0.939 (0.039) | 4.102 (1.637) | 1.904 (1.096) | 2.135 (1.086) | 4.48 | 9.92 |
| | db8 | $(4,5,8)$ | 0.940 (0.036) | 4.107 (1.572) | 1.821 (1.073) | 2.058 (1.272) | 4.11 | 9.26 |
| | db16 | $(5,5,8)$ | 0.940 (0.037) | 4.175 (1.371) | 1.880 (1.025) | **1.968** (1.183) | 3.98 | 9.92 |
| | db2- | $(2,-,8)$ | 0.761 (0.032) | 14.611 (4.810) | 7.344 (2.609) | 6.511 (1.864) | 3.97 | 9.77 |
| | db4- | $(3,-,8)$ | 0.930 (0.041) | 4.661 (1.592) | 2.050 (0.768) | 2.429 (1.129) | **3.97** | 9.35 |
| | db8- | $(4,-,8)$ | 0.938 (0.042) | 4.205 (2.303) | 1.990 (1.455) | 2.155 (1.315) | 3.98 | 8.99 |
| | db16- | $(5,-,8)$ | 0.939 (0.035) | 4.058 (1.517) | 1.798 (0.881) | 1.960 (1.007) | 3.98 | **8.79** |
| | Ronneberger et al. (2015) | $-$ | **0.952** (0.040) | 4.369 (4.913) | $-$ | $-$ | 7.76 | 38.9 |
| | Nikolov et al. (2018) | $-$ | 0.948 (0.038) | **3.584** (1.397) | $-$ | $-$ | 6.16 | 40.7 |
| **Prostate** | db1 | $(1,5,6)$ | 0.931 (0.032) | 5.328 (2.587) | 2.369 (1.269) | 2.407 (0.803) | 8.38 | 11.2 |
| | db2 | $(2,5,6)$ | 0.930 (0.035) | 5.450 (2.647) | 2.148 (1.360) | 2.120 (0.788) | 5.07 | 9.66 |
| | db4 | $(3,5,6)$ | 0.935 (0.032) | 5.333 (2.572) | **2.026** (1.242) | **2.005** (0.894) | 4.17 | 8.64 |
| | db8 | $(4,5,6)$ | 0.931 (0.032) | **5.323** (2.584) | 2.147 (1.381) | 2.019 (0.873) | 3.87 | 7.59 |
| | db16 | $(5,5,6)$ | 0.924 (0.040) | 5.583 (2.665) | 2.218 (1.280) | 2.197 (0.994) | 3.74 | 7.65 |
| | db2- | $(2,-,6)$ | 0.779 (0.025) | 14.584 (4.487) | 6.072 (1.937) | 6.577 (1.998) | 3.73 | 8.22 |
| | db4- | $(3,-,6)$ | 0.921 (0.037) | 6.040 (3.264) | 2.353 (1.316) | 2.328 (1.046) | 3.73 | 7.84 |
| | db8- | $(4,-,6)$ | 0.923 (0.041) | 6.082 (3.380) | 2.334 (1.485) | 2.328 (1.008) | **3.73** | 7.79 |
| | db16- | $(5,-,6)$ | 0.928 (0.032) | 5.499 (2.901) | 2.177 (1.273) | 2.078 (0.784) | 3.74 | **7.56** |
| | Ronneberger et al. (2015) | $-$ | 0.932 (0.047) | 5.673 (2.402) | $-$ | $-$ | 7.76 | 37.2 |
| | Nikolov et al. (2018) | $-$ | **0.937** (0.030) | 5.475 (2.380) | $-$ | $-$ | 5.49 | 32.4 |

resulting in curves with subtle spurious curvature. Such geometry is not (and cannot be) present in the binary ground-truth mask due its discrete nature.

**Prostate central gland**   The wavelet models produce accurate predictions and are on par with the U-Nets. Our models perform slightly better in terms of Hausdorff distance. While all models produce accurate predictions for most examples, there are instances where both the wavelet models and the baseline U-Nets fail to produce accurate predictions; see the last two columns of Figure 5. In these examples, the detail coefficients associated to parts of the curve with high curvature are too small in magnitude to be accurately predicted. While detail coefficients of such small magnitude were less relevant in the latter two examples, they are important for the prostate.

**Ablation study (no detail coefficients)**   To demonstrate the importance of the detail coefficients, we have trained models without them, i.e., without the skip-connections. In this set up, we do not supervise the predictions on the lowest resolution level during training. The results demonstrate, as expected, that the lower-order wavelets db1, db2 perform significantly worse without detail coefficients. In fact, for db1 our model failed to produce any sensible approximations that can be evaluated and were therefore omitted. A small drop in performance is observed for db4. In general, for db16 there is not much gain, with respect to accuracy, memory-footprint and inference time, to explicitly model the detail coefficients.

## 6   CONCLUSION

We have introduced a novel method to model boundaries of two dimensional simply connected domains using wavelets and MRAs. In effect this allows for subpixel segmentations. The efficacy of the method has been demonstrated by modeling the boundaries of hypercloids (toy-example), spleen and prostate, demonstrating that the results are on par with typical U-Nets, yielding up to five times faster inference speed.

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

## A    THEORETICAL RESULTS

In this section we prove theoretical results regarding the computation of wavelet decompositions. In addition, we provide all the implementation details.

### A.1    EXPLICIT EXPRESSION FOR $a_{jk}$

To address the issue that periodic signals are not contained in $L^2(\mathbb{R})$, we consider the cut-off $\tilde{\gamma} := \gamma \mathbf{1}_{[-l,l]}$. In this section we derive an explicit formula for $\langle \tilde{\gamma}, \varphi_{jk} \rangle$ by exploiting the periodicity of $\gamma$. In particular, we will quantify the claim that (scaled) sample values of $\gamma$ may be used as approximation coefficients. For this purpose, we first review some facts about the scaling equation. If we take the Fourier transform of both sides of the scaling equation, we obtain the relation $\hat{\varphi}(\xi) = H(\frac{\xi}{2})\hat{\varphi}(\frac{\xi}{2})$ for $\xi \in \mathbb{R}$, where $H(\xi) = \frac{1}{\sqrt{2}} \sum_{k \in \mathbb{Z}} h_k e^{-i2\pi\xi k}$. The map $H$ is referred to as the *refinement mask* associated to $h$. Throughout this paper we assume that $h$ is finite, in which case $\varphi$ has compact support and $H$ is a trigonometric polynomial of period 1. It is beyond the scope of this paper to discuss the properties of $H$ in detail and refer the reader to Pereyra & Ward (2012); Montefusco & Puccio (2014). Here we only need the following result. Under suitable conditions, one may iterate the equation for $\hat{\varphi}$ and show that $\hat{\varphi}(\xi) = \prod_{k=1}^{\infty} H(\frac{\xi}{2^k})$ for all $\xi \in \mathbb{R}$.

**Lemma A.1** (Initialization approximation coefficients). *Let $h \in \ell^2(\mathbb{Z})$ be a low-pass-filter defining a MRA on $L^2(\mathbb{R})$ and $H$ the associated refinement mask. Assume $h$ is non-zero for only a finite number of coefficients, $\mathrm{supp}(\varphi) \subset [0, \beta]$ for some $\beta > 0$, and $\varphi$ is bounded. Furthermore, suppose that $\hat{\varphi}(\xi) = \prod_{n=1}^{\infty} H(\frac{\xi}{2^n})$ for all $\xi \in \mathbb{R}$. If $\gamma \in C_{per}^2([0, l])$ is a l-periodic map with Fourier coefficients $(\gamma_m)_{m \in \mathbb{Z}}$, then*

$$\langle \tilde{\gamma}, \varphi_{jk} \rangle = 2^{-\frac{j}{2}} \sum_{m \in \mathbb{Z}} \gamma_m e^{i\omega(l)m\frac{k}{2^j}} \prod_{n=1}^{\infty} H\left(-\frac{m}{l2^{j+n}}\right), \tag{1}$$

*where $\omega(l) := \frac{2\pi}{l}$ is the angular frequency of $\gamma$, for any $j \in \mathbb{Z}$ and $k \in \{\lceil -2^j l \rceil, \ldots, \lfloor 2^j l - \beta \rfloor\}$.*

*Proof.* Let $j \in \mathbb{Z}$ and $k \in \{\lceil -2^j l \rceil, \ldots, \lfloor 2^j l - \beta \rfloor\}$ be arbitrary. A change of variables shows that

$$\langle \tilde{\gamma}, \varphi_{jk} \rangle = 2^{-\frac{j}{2}} \int_{[0,\beta]} \tilde{\gamma}\left(2^{-j}(t+k)\right) \varphi(t) \, \mathrm{d}t, \quad k \in \mathbb{Z},$$

since $\mathrm{supp}(\varphi) \subset [0, \beta]$. In particular, note that $2^{-j}(t+k) \in [-l, l]$ for $t \in [0, \beta]$, since $k \in \{\lceil -2^j l \rceil, \ldots, \lfloor 2^j l - \beta \rfloor\}$. Therefore, we may plug in the Fourier expansion for $\gamma$ and compute

$$\int_{[0,\beta]} \tilde{\gamma}\left(2^{-j}(t+k)\right) \varphi(t) \, \mathrm{d}t = \int_{[0,\beta]} \sum_{m \in \mathbb{Z}} \gamma_m e^{i\omega(l)m\frac{t+k}{2^j}} \varphi(t) \, \mathrm{d}t.$$

Next, note that that series inside the integral converges pointwise to $\gamma\left(2^{-j}(t+k)\right)\varphi(t)$ on $[0, \beta]$. Furthermore, the partial sums can be bounded from above on $[0, \beta]$ by a constant, since $\gamma \in C_{\mathrm{per}}^2([0, l])$ and $\varphi$ is bounded. Therefore, we may interchange the order of summation and integration by the Dominated Convergence Theorem:

$$\int_{[0,\beta]} \sum_{m \in \mathbb{Z}} \gamma_m e^{i\omega(l)m\frac{t+k}{2^j}} \varphi(t) \, \mathrm{d}t = \sum_{m \in \mathbb{Z}} \gamma_m e^{i\omega(l)m\frac{k}{2^j}} \int_{[0,\beta]} e^{i\omega(l)m\frac{t}{2^j}} \varphi(t) \, \mathrm{d}t.$$

Finally, changing the domain of integration to $\mathbb{R}$ again, we see that

$$\sum_{m \in \mathbb{Z}} \gamma_m e^{i\omega(l)m\frac{k}{2^j}} \int_{[0,\beta]} e^{i\omega(l)m\frac{t}{2^j}} \varphi(t) \, \mathrm{d}t = \sum_{m \in \mathbb{Z}} \gamma_m e^{i\omega(l)m\frac{k}{2^j}} \hat{\varphi}\left(-\frac{m}{l2^j}\right).$$

The stated result now follows from the assumption that $\hat{\varphi}(\xi) = \prod_{k=1}^{\infty} H(\frac{\xi}{2^k})$ for any $\xi \in \mathbb{R}$. $\square$

**Remark A.2.** *The bounds $\lceil -2^j l \rceil$ and $\lfloor 2^j l - \beta \rfloor$ are the smallest and largest integer, respectively, for which $2^{-j}(t+k) \in [-l, l]$ for all $t \in [0, \beta]$. The bounds on $k$ are somewhat artificial, however, since the argument may be repeated for any truncation of $\gamma$ on $[-nl, nl]$, where $n \in \mathbb{N}$. This observation is reflected in the righthand-side of (1), which is well-defined for all $k \in \mathbb{Z}$ (but not squared-summable). The reason for choosing this particular truncation is to identify a minimal number of approximation coefficients needed to cover the full signal $\gamma$.*

The requirement that $\hat{\varphi}(\xi) = \prod_{k=1}^{\infty} H(\frac{\xi}{2^k})$ for $\xi \in \mathbb{R}$ is satisfied for all MRA's considered in this paper. In particular, we remark that $\hat{\varphi}$ is continuous at $\xi = 0$ and $\hat{\varphi}(0) = 1$. It follows from these observations and Lemma A.1 that $a_{jk}(\tilde{\gamma}) = a_{jk}(\gamma) \approx 2^{-\frac{j}{2}} \gamma(k2^{-j})$ for $j$ sufficiently large and $k$ constrained to $\{\lceil -2^j l \rceil, \ldots, \lfloor 2^j l - \beta \rfloor\}$.

## A.2 PYRAMID ALGORITHM - IMPLEMENTATION DETAILS

In this section we provide the computational details for how to compute approximation and detail coefficients using the Discrete Fourier Transform (DFT). In the following arguments all sequences are assumed to be two-sided. We will frequently abuse notation and write that a finite sequence is an element in $\mathbb{C}^n$ or $\mathbb{R}^n$. What we actually mean by this, is that we have a two-sided sequence of length $n$ which can be embedded in $\ell^2(\mathbb{Z})$ by appropriately padding with zeros. In such a situation we will explicitly specify the indices of the sequence so that the intended ordering is clear. Similarly, all operators in this section are implicitly assumed to be defined on the associated (two-sided) sequence spaces, even though we may write that they are defined on $\mathbb{C}^n$ or $\mathbb{R}^n$.

Let $a = (a_k)_{k=-N}^{N-1} \in \mathbb{R}^{2N}$ and $g = (g_k)_{k=1-M}^{M-1} \in \mathbb{R}^{2M-1}$ be arbitrary. Here $g$ may be interpreted as a high-pass-filter of length $M$ with $N \geq M \geq 2$. Similarly, we may think of $a$ as the approximation coefficients of a 1-periodic signal at a specific resolution level $j$, with $N = 2^{j-1}$, as explained in Section 2.3. In this section, however, we will not emphasize these interpretations, e.g., write $a_{jk}$ instead of $a_k$, to avoid clutter in the notation. Observe that the righthand-side of (1) is a $2^j$-periodic sequence for 1-periodic signals. Therefore, to properly deal with "boundary terms", we will use the $2N$-periodic extension $\tilde{a} \in \mathbb{R}^{\mathbb{Z}}$ of $a$ to evaluate discrete convolutions.

**Convolution and multiplication of trigonometric polynomials** To compute the detail and approximation coefficients, we need to evaluate expressions of the form

$$(\tilde{a} * g)_k = \sum_{\substack{k_1+k_2=k \\ |k_2| \leq M-1 \\ k_1 \in \mathbb{Z}}} \tilde{a}_{k_1} g_{k_2} = \sum_{|k_2| \leq M-1} \tilde{a}_{k-k_2} g_{k_2} \qquad (2)$$

for $-N \leq k \leq N-1$. Note that although $\tilde{a}$ is an infinite $2N$-periodic sequence, the series in (2) contains only a finite number of nonzero-terms, since $g_{k_2} = 0$ for $|k_2| \geq M$. Furthermore, for $-N \leq k \leq N-1$, we do not need the full periodic extension $\tilde{a}$, but only a partial (finite) extension $P_K(a)$, where $P_K : \mathbb{C}^{2N} \to \mathbb{C}^K$ is defined by

$$(P_K(a))_k := \begin{cases} a_{k+2N} & 1-N-M \leq k \leq -N-1, \\ a_k & -N \leq k \leq N-1, \\ a_{k-2N} & N \leq k \leq N+M-2 \end{cases}$$

and $K := 2(N+M-1)$. That is, $(\tilde{a} * g)_k = (P_K(a) * g)_k$ for $-N \leq k \leq N-1$.

We use standard arguments to compute $P_K(a) * g$. Namely, we interpret $P_K(a) * g$ as the Fourier coefficients of $uv$, where $u, v : \mathbb{R} \to \mathbb{C}$ are the trigonometric polynomials defined by

$$u(\theta) = \sum_{k=-\frac{K}{2}}^{\frac{K}{2}-1} (P_K(a))_k e^{ik\theta}, \quad v(\theta) = \sum_{k=1-M}^{M-1} g_k e^{ik\theta}.$$

The product $uv$ is a trigonometric polynomial with $\tilde{K} := K + 2(M-1)$ non-zero coefficients corresponding to terms of order $-\frac{\tilde{K}}{2} \leq k \leq \frac{\tilde{K}}{2} - 1$. The coefficients of $uv$ can be characterized by evaluating it on $\tilde{K}$ distinct points in $\mathbb{C}$. After fixing $\tilde{K}$ such points, we may go back and forth between value and coefficient representations of $u, v$ and $uv$ using the isomorphism defined by the evaluation operator.

**Evaluation at the roots of unity** We evaluate $u$ and $v$, in the complex variable $z = e^{i\theta}$, at the $\tilde{K}$-th roots of unity. To do this, we first extend $P_K(a)$ and $g$ to sequences in $\mathbb{C}^{\tilde{K}}$ by padding with zeros. More precisely, define $Z_{\tilde{K}}^{\text{even}} : \mathbb{C}^K \to \mathbb{C}^{\tilde{K}}$ by $(Z_{\tilde{K}}^{\text{even}}(b))_k = b_k$ for $-\frac{K}{2} \leq k \leq \frac{K}{2} - 1$ and zero for $-\frac{\tilde{K}}{2} \leq k < -\frac{K}{2}$ and $\frac{K}{2} \leq k < \frac{\tilde{K}}{2}$. Similarly, define $Z_M^{\text{odd}} : \mathbb{C}^{2M-1} \to \mathbb{C}^{\tilde{K}}$ by $(Z_M^{\text{odd}}(b))_k = b_k$ for $|k| < M$ and zero for $-\frac{\tilde{K}}{2} \leq k \leq -M$ and $M \leq k < \frac{\tilde{K}}{2}$. We can now evaluate $u$ and $v$ at the $\tilde{K}$-th roots of unity by computing $\mathbf{DFT}_{\tilde{K}} \circ S_{\tilde{K}} \circ Z_{\tilde{K}}^{\text{even}} \circ P_K(a)$ and $\hat{\mathbf{G}}_{\tilde{K}} := \mathbf{DFT}_{\tilde{K}} \circ S_{\tilde{K}} \circ Z_M^{\text{odd}}(g)$, respectively, where $S_{\tilde{K}} : \mathbb{C}^{\tilde{K}} \to \mathbb{C}^{\tilde{K}}$ is defined by

$$(S_{\tilde{K}} b)_k := \begin{cases} b_k, & 0 \leq k \leq \frac{\tilde{K}}{2} - 1, \\ b_{k-\tilde{K}}, & \frac{\tilde{K}}{2} \leq k \leq \tilde{K} - 1. \end{cases}$$

Consequently, $uv$ can be evaluated at the $\tilde{K}$-th roots of unity by taking the element-wise product of the latter two vectors. Finally, the desired coefficients $(P_K(a) * g)_{k=-N}^{N-1}$ are obtained by going back to coefficient space using the inverse DFT , i.e.,

$$(P_K(a) * g)_{k=-N}^{N-1} = \Pi_N S_{\tilde{K}}^{-1} \mathbf{DFT}_{\tilde{K}}^{-1} \left( \hat{\mathbf{G}}_{\tilde{K}} \odot \mathbf{DFT}_{\tilde{K}} \circ S_{\tilde{K}} \circ Z_{\tilde{K}}^{\text{even}} \circ P_K(a) \right).$$

Here $\odot$ denotes the Hadamard-product and $\Pi_N : \mathbb{C}^{\tilde{K}} \to \mathbb{C}^{2N}$ is the truncation operator defined by $\Pi_N(b) := (b_k)_{k=-N}^{N-1}$.

## B PREPROCESSING

In this section we provide the details of our preprocessing steps.

### B.1 TRUNCATION FOURIER COEFFICIENTS

The magnitude of the approximated Fourier coefficients will typically stagnate and stay constant (approximately) beyond some critical order, since all computations are performed in finite (single) precision. We locate this critical order $m_0^*(s) \in \mathbb{N}$ for each component $s \in \{1, 2\}$, if present, by iteratively fitting the best line, in the least squares sense, through the points $\left\{ \left( m, \left\| (|[\tilde{\gamma}_{\tilde{m}}]_s|)_{\tilde{m}=m_0}^m \right\|_1 \right) : m_0 \leq m \leq N - 1 \right\}$ for $1 \leq m_0 \leq N - 1$. We iterate this process until the residual is below a prescribed threshold $\delta_N > 0$. In practice, we set $\delta_N = 0.1$. The Fourier coefficients with index strictly larger than $m_0^*(s)$ are set to zero.

### B.2 CONSISTENT PARAMETERIZATIONS

To have consistent parameterizations we enforce that all contours start at angle zero at time zero relative to the midpoint $c = (c_1, c_2) \in \mathbb{R}^2$ of the region of interest $R$. This is accomplished by exploiting the Fourier representation of the curve. More precisely, let $\tilde{\gamma} := t \mapsto \sum_{|m| \leq N-1} \tilde{\gamma}_m e^{i\omega(l)mt}$, where $\omega(l) = \frac{2\pi}{l}$, be the contour with Fourier coefficients $\Gamma := (\tilde{\gamma}_m)_{m=1-N}^{N-1}$. The midpoint $c$ of the region enclosed by $\tilde{\gamma}$ is given by

$$c_s = \frac{1}{\lambda(R)} \int_R u_s \, d\lambda(u_1, u_2) = (-1)^s \frac{([\Gamma]_1 * [\Gamma]_2 * [\Gamma']_s)_0}{([\Gamma]_1 * [\Gamma']_2)_0}, \quad s \in \{1, 2\} \tag{3}$$

by Green's Theorem. Here $\lambda$ denotes the Lebesgue measure on $\mathbb{R}^2$, $[\Gamma]_s$ are the Fourier coefficients of $[\tilde{\gamma}]_s$, and $(\Gamma')_m := im\omega(l)\gamma_m$ for $|m| \leq N - 1$. We can now compute the desired parameterization by determining $\tau \in [0, l]$ such that $\arccos \left( \frac{[\tilde{\gamma}(-\tau) - c]_1}{\|\tilde{\gamma}(-\tau) - c\|_2} \right) \approx 0$ and defining $\gamma(t) := \tilde{\gamma}(t - \tau)$. While $\tau$ can be easily found using Newton's method, it suffices in practice to simply re-order $y$ from the start, before computing the Fourier coefficients. More precisely, we first define a shift $\tilde{y}$ of $y$ by $\tilde{y}_k := y_{k + k^* \bmod n_p}$ for $0 \leq k \leq n_p - 1$, where $k^* := \text{argmin} \left\{ \arccos \left( \frac{[y_k - c]_1}{\|y_k - c\|_2} \right) \right\}_{k=0}^{n_p - 1}$.

## C U-NET ARCHITECTURES (BASELINES)

In this section we describe our modifications to the networks presented in Ronneberger et al. (2015) and Nikolov et al. (2018). The goal of the modifications is to match the parameter count with our networks (approximately), while maintaining the structure and idea's presented in the original papers as much as possible. Our modifications to the original U-Net in Ronneberger et al. (2015) is minimal: we use 32 filters in the first layer instead of 64 and use group normalization with four groups. Our modifications to the network in Nikolov et al. (2018) are as follows. We use residual blocks of length two (instead of three), we use a bottleneck with 256 channels (instead of 1024), and we do not use 1d convolutions in the "third" direction (to keep our model 2d). Furthermore, we use an encoder and decoder of length six. The number of filters used in the encoder, from top to bottom, is 32, 32, 64, 64, 128, and 128, respectively. The number of filters used in the decoder, from bottom to top, is 128, 128, 64, 64, 64 and 64, respectively.

# D FIGURES

In this section we provide additional visualizations of the statistics, wavelet decompositions and predicted contours.

## D.1 TOY EXAMPLE

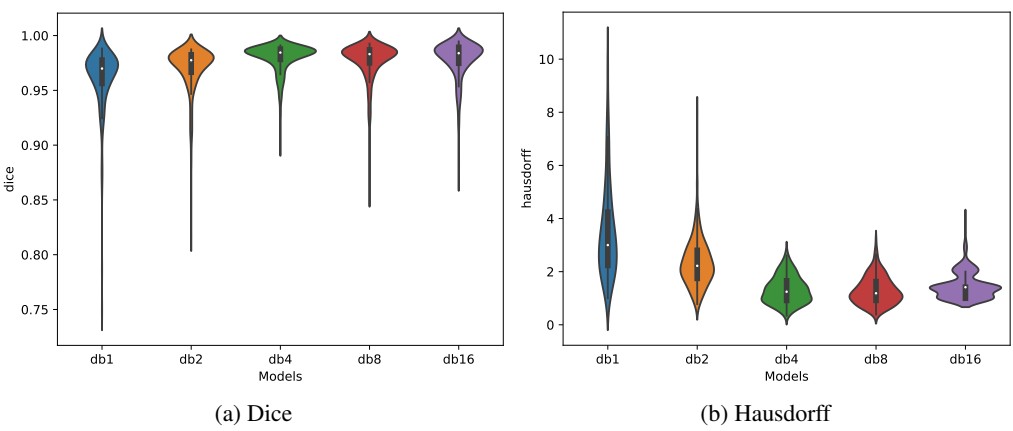

(a) Dice

(b) Hausdorff

Figure 6: Boxplots and visualization of approximate densities for the dice scores and Hausdorff distances for the toy problem.

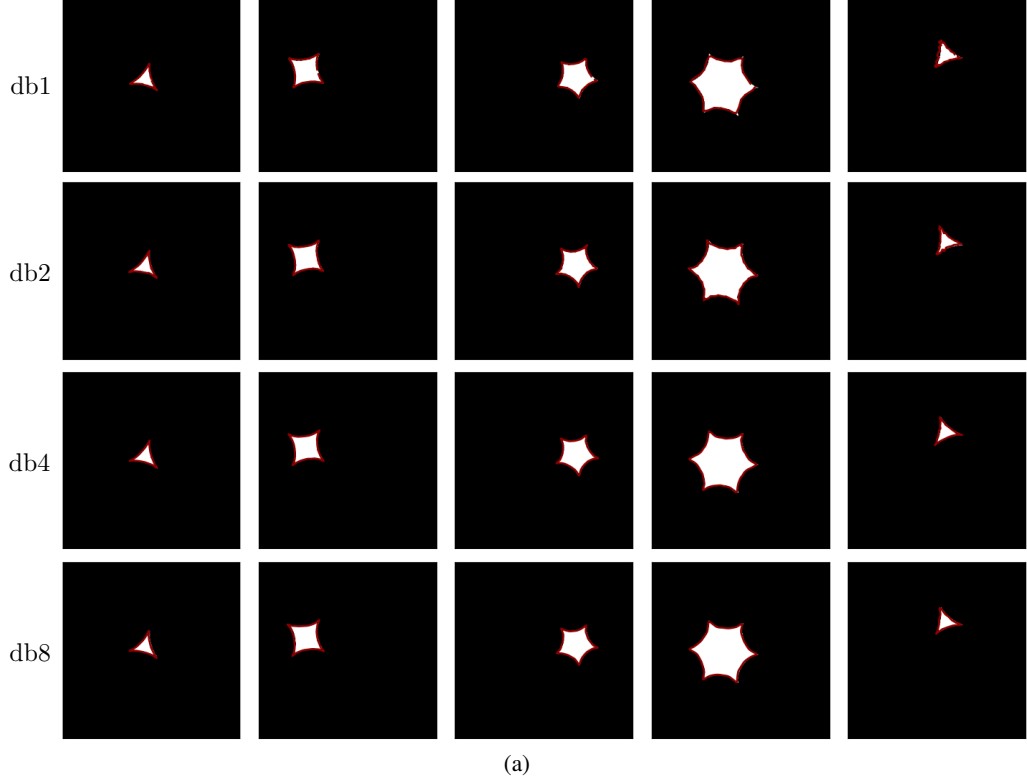

(a)

Figure 7: Predicted curve (in red) for a binary mask (in white). In order to visualize the predicted curve without too much clutter we have not depicted the ground-truth contour.

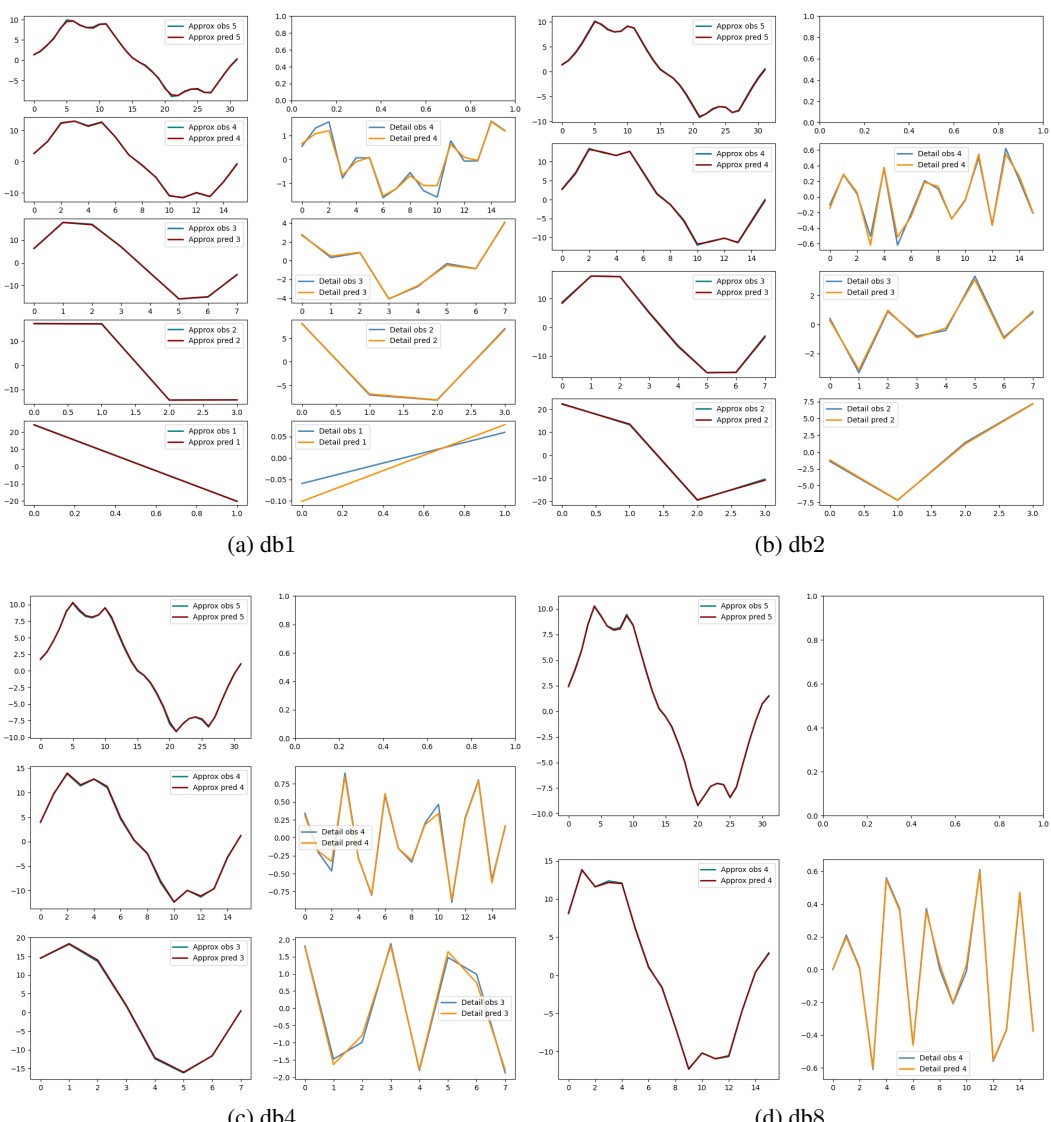

Figure 8: Predicted and observed wavelet decompositions of the first component of the contour depicted in the fourth column in Figure 7.

### D.1.1 SPLEEN

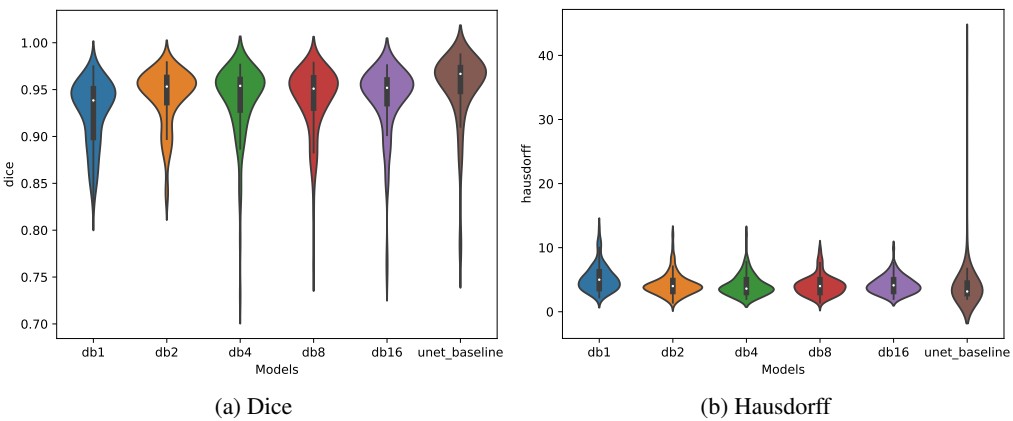

(a) Dice                    (b) Hausdorff

Figure 9: Boxplots and visualization of approximate densities for the dice scores and Hausdorff distances for the spleen.

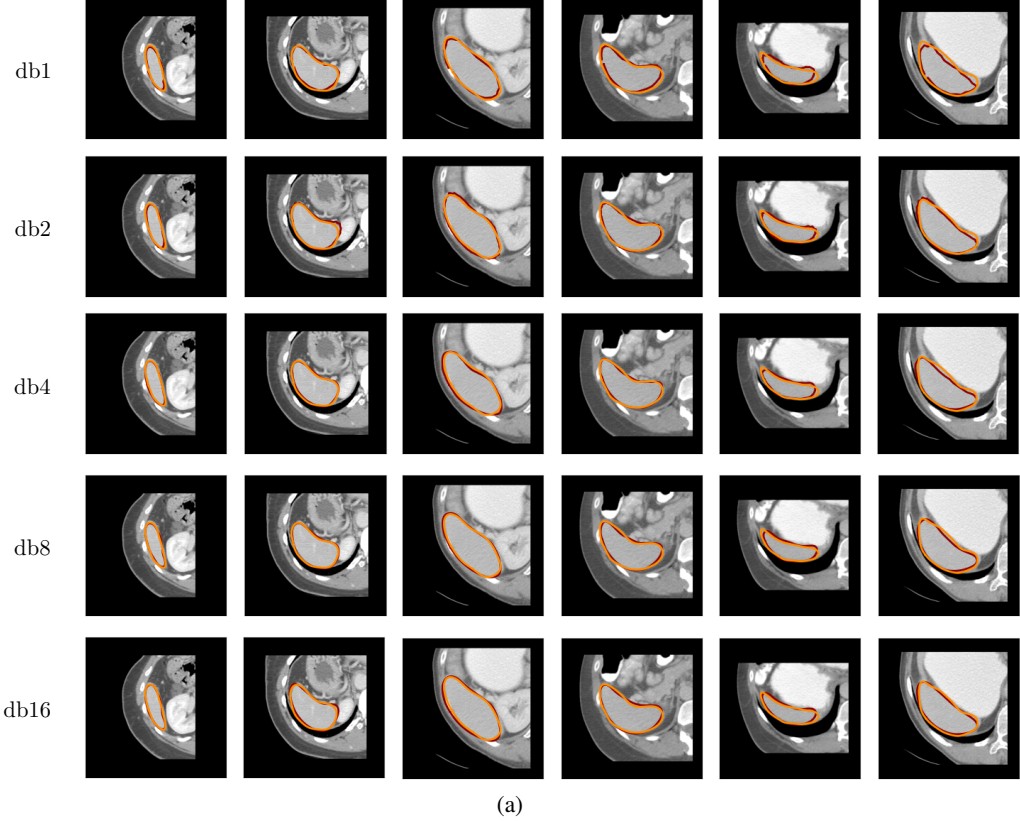

(a)

Figure 10: Predicted and observed boundaries colored in red and orange, respectively, of the spleen. The last two columns correspond to a few "hard" examples.

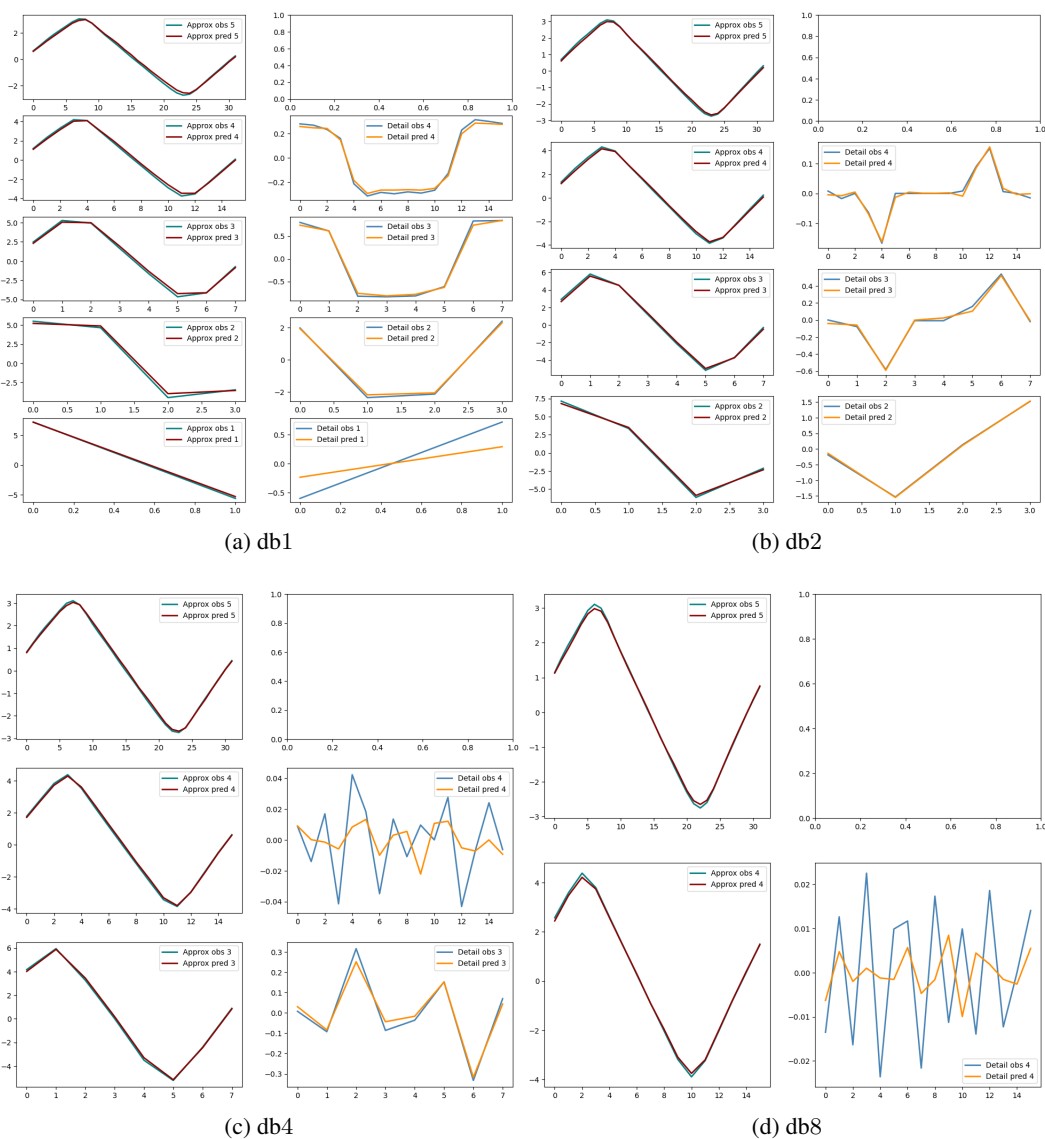

Figure 11: Predicted and observed wavelet decompositions of the first component of the contour depicted in the first column in Figure 10. For db4 and db8 the detail coefficients on higher resolution levels are too small to be accurately predicted, but have little impact on the accuracy of the final approximation.

### D.1.2 PROSTATE

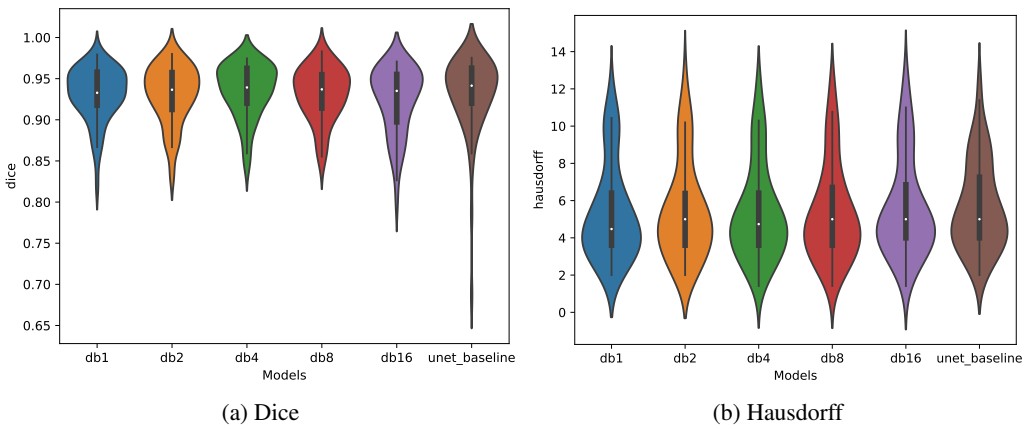

(a) Dice       (b) Hausdorff

Figure 12: Boxplots and visualization of approximate densities for the dice scores and Hausdorff distances for the prostate.

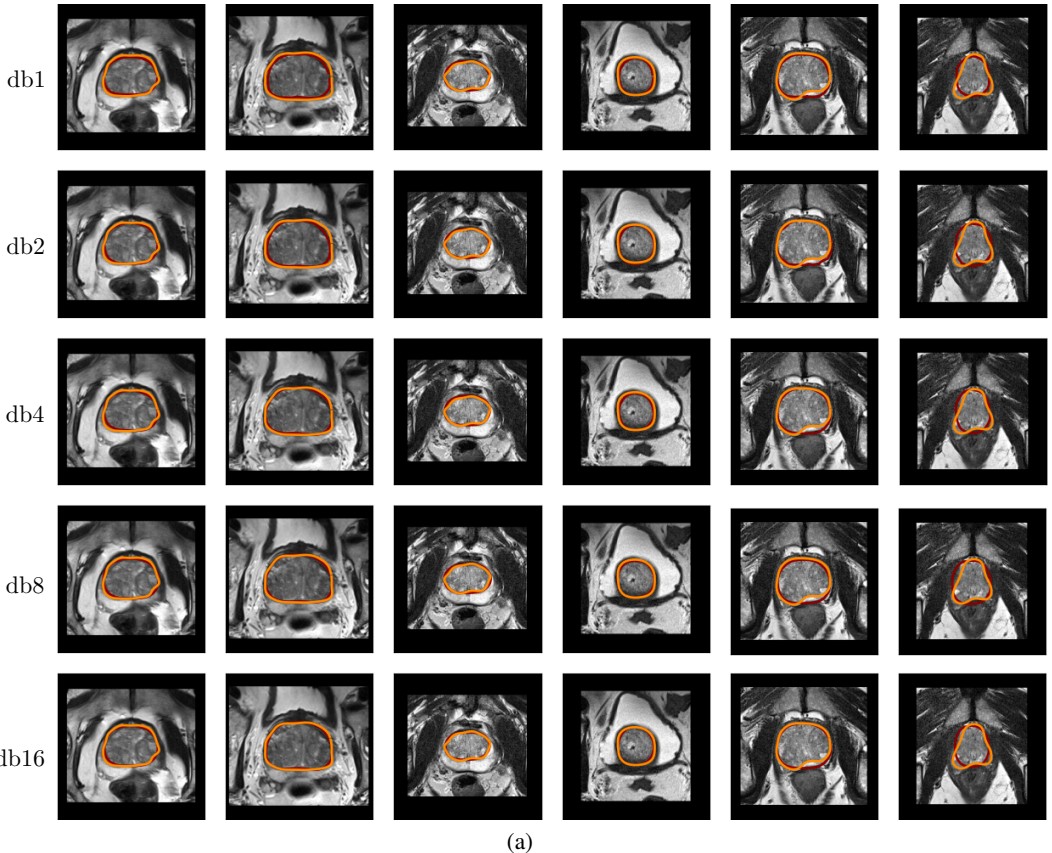

(a)

Figure 13: Predicted and observed boundaries colored in red and orange, respectively, of the prostate. The last two columns correspond to "hard" examples.

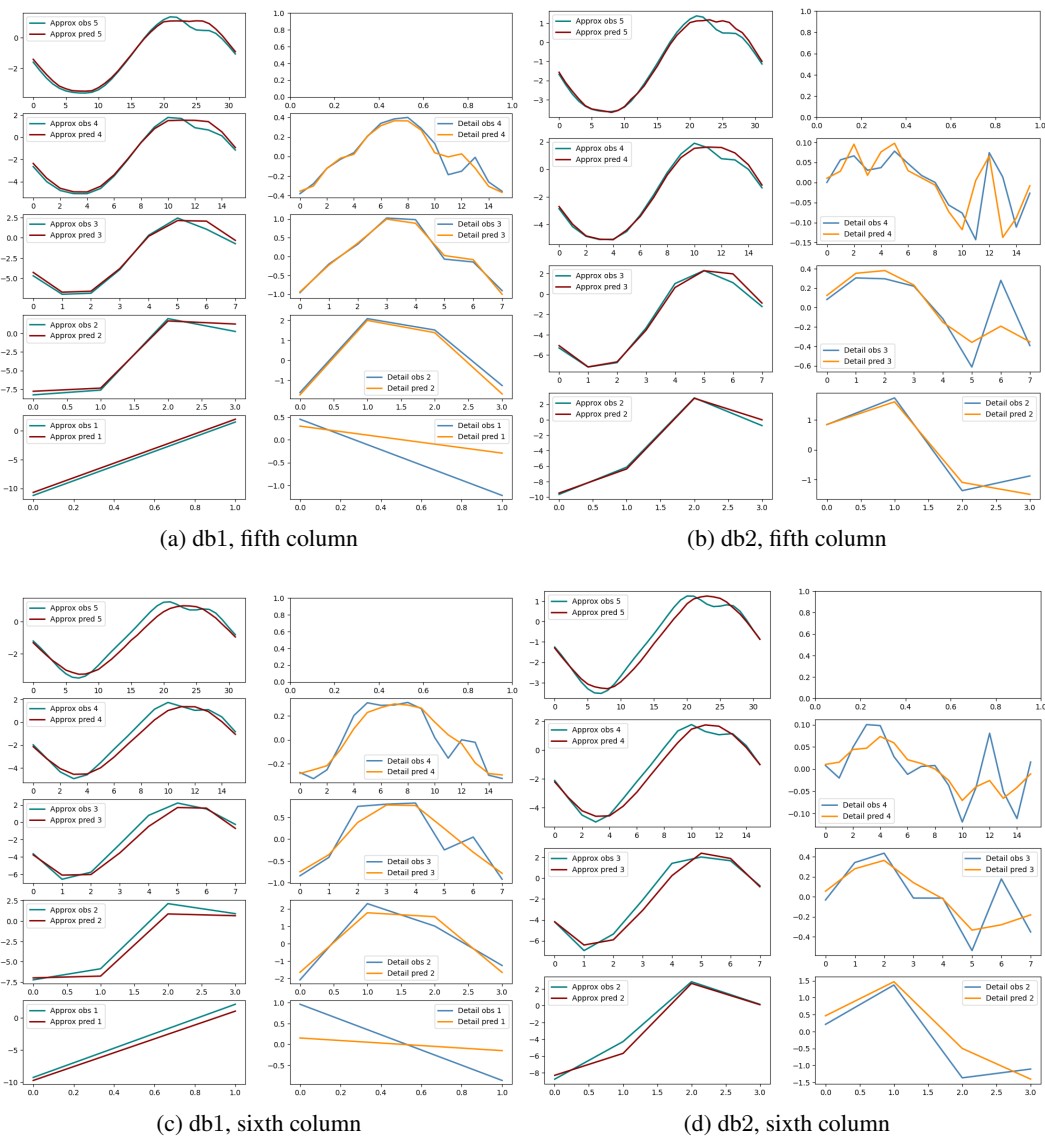

(a) db1, fifth column

(b) db2, fifth column

(c) db1, sixth column

(d) db2, sixth column

Figure 14: Predicted and observed wavelet decompositions of the first component of the hard examples depicted in  (a),  (b) the fifth column and  (c),  (d) the sixth column of Figure 13.

