# OpenReview forum: "Subpixel object segmentation using wavelets and multiresolution analysis"
_ICLR.cc/2022/Conference — ICLR 2022 Submitted_

### Official Review · Reviewer_xHYe · 2021-10-21

**Correctness:** 2
**Technical Novelty And Significance:** 3
**Empirical Novelty And Significance:** 3
**Recommendation:** 6
**Confidence:** 4

**Main Review:**

The article is well written although it is complex to follow.
The mathematical introduction is quite dense. Besides, there are some functions that are not defined (such as L(Z) and l(Z)), which makes it difficult to understand the article for those who do not know how Wavelets work. The state of the art is quite good given the space limitations although it would have been nice to go a bit deeper into the differences with respect to previous proposals (apart from the subpixel resolution).

 However, the main weaknesses of the paper are:
1.- Although there are other similar approaches (and even with sub-pixel level segmentation) using wavelets the authors compare with a generic segmentation approach based on a classical encoder-decoder structure.

2.- From the approach with which they compare, they fix the hyperparameters but do not justify why (and do not adjust them to improve the results).

3.- In the comparison, the authors do look for the best hyperparameters of their approximation by comparing the best one with the (prefixed) approximation of the generic method. This dilutes the "power" of the results obtained.

**Summary Of The Paper:**

In this paper the authors propose a mixed approach to image segmentation using CNNs and wavelets. Although this approach is not new, to the best of the reviewer's knowledge, it is the first time it has been applied to medical image segmentation. It basically consists of applying an encoder-decoder structure in which the encoder is a CNN and the decoder is based on a traditional (Pyramid Algorithm) Wavelet method. To validate the proposed method, they compare their results with a general and complete CNN method. The results demonstrate that it could be a powerful approximation: fast, accurate and easy to train.

**Summary Of The Review:**

Pros:
The paper is well written and the proposal is very interesting from the point of view of the medical image segmentation.

Cons:
The mathematical explanation must be improved to help the understandability of the paper.
The validation of the proposal must be compared with other wavelet proposals.
Authors should be fairer when adjusting the hyperparameters (not limited to the proposed model).

---

> ### Author Response · Authors · 2021-11-22
> **Thank you for your feedback!**
>
> Thank you for your feedback! Your feedback was very useful for improving our paper. We have provided detailed answers to your questions below.
>
> **Article is complex to follow. The mathematical explanation must be improved to help the understandability of the paper.**
>
> We have added schematic pictures to visualize the mathematical content in Section 2. In addition, whenever possible, we have omitted or complemented mathematical details with illustrational figures. In particular, we have added a schematic picture of the Pyramid Algorithm,  which forms the basis of our method. In addition, we have added a detailed picture of the architecture of our network. In general, we have added figures to help the reader visualize the complete pipeline.
>
> **Undefined notation which makes it difficult to understand the article for those who do not know how Wavelets work.**
>
> We have tried to use standard notation from linear algebra and functional analysis only, which is not specific to wavelets. However, we realize now that this notation may not be as mainstream as initially anticipated. We have clarified unknown notation and whenever possible completely avoided too precise descriptions.
>
> **It would have been nice to go a bit deeper into the differences with respect to previous proposals (apart from the subpixel resolution).**
>
> We agree with the reviewer that it is unfortunate that we have not been able to delve more deeply into other methods, such as active contours and snaking methods. However, due to limited space, we were not able to perform such an analysis.
>
> **There are other similar approaches (and even with sub-pixel level segmentation) using wavelets, but the authors compare with a generic segmentation approach based on a classical encoder-decoder structure.**
>
> We agree with the reviewer that it would have been more appropropriate to compare our method with other contour-based methods. However, due to time constraints, we will not be able to provide such a comparison.
>
> We are unfortunately not aware of approaches similar to ours using wavelets. We are aware of methods incorporating wavelet decompositions of 2d images into CNNs to (i) perform classification (ii) perform classical image segmentation (iii) perform generative modeling. In (i) and (ii) the focus seems to be on learning improved features from images to perform the task at hand, while (iii) focuses on generating 2d images. In all cases, the purpose of the wavelet decompositions are fundamentally different than ours. We have not been able to find papers which focus on the task of parameterizing planar curves using wavelet decompositions in a deep learning context.
>
> **From the approach with which they compare, they fix the hyperparameters but do not justify why (and do not adjust them to improve the results). In the comparison, the authors do look for the best hyperparameters of their approximation by comparing the best one with the (prefixed) approximation of the generic method. This dilutes the "power" of the
> results obtained.**
>
> The main purpose of the experiments was to investigate the dependence of our method on the chosen basis, i.e., the type of wavelet. In doing so, we explained in Section 5 why many other hyperparameters, most notably the resolution levels j_0 and j_1 and j_2 need not be varied and can be fixed from the start. From this perspective, we did not tune or look for optimal hyperparameters in our models. We simply tried out different wavelet bases to investigate their effect. We have emphasized this more clearly now at the beginning of Section 5.
>
> We agree with the reviewer that the parameters for the U-Net are not appropriately discussed. The configuration of the network was mainly motivated by the desire to have a network of “similar size” as ours while still achieving accurate results with respect to its own objective (predicting binary masks). The purpose for including the U-Net was only to give a sense of the performance of a standard framework of similar size with an objective close to ours. We have added an additional experiment using a more advanced and refined U-Net for comparison and included the above explanation to Section 5.
>
> **Authors should be fairer when adjusting the hyperparameters (not limited to the proposed model). The validation of the proposal must be compared with other wavelet proposals.**
>
> We refer the reviewer to the previous two points.

---

### Official Review · Reviewer_MBzp · 2021-11-02

**Correctness:** 2
**Technical Novelty And Significance:** 2
**Empirical Novelty And Significance:** 2
**Recommendation:** 3
**Confidence:** 4

**Main Review:**

Strength: the use of wavelets and mulit-resolution analysis could be new.
Weakness: the paper is not well presented.
1) Section 2 could be simplified.
2) The motivation of the paper is not clear. Why it is subpixel?
3) The authors shall highlight their contributions and clarify the difference from the state-of-the-art algorithms.
4) It is not clear if the use of wavelets and multi-resolution analysis is a significant contribution or not.
5) I suggest the authors to draw a diagram of their network.
6) The improvement seems marginal compared with U-net
7) There are many methods for segmentation that shall be cited and compared.

**Summary Of The Paper:**

This paper presents a method for segmentation using wavelets and Multi Resolution Analysis.
 But the presentation of the paper needs to be improved to highlight the technical novelty and difference from the state-of-the-art algorihtms.



**Summary Of The Review:**

Given the weakness stated above, I felt that the paper is not strong enough.

---

> ### Author Response · Authors · 2021-11-22
> **Thank you for your feedback!**
>
> Thank you for your feedback! We have addressed your questions and concerns in our paper to improve the presentation. In addition, we have provided point-to-point answers to your questions / comments below.
>
> **Section 2 could be simplified**
>
> We have added schematic pictures to visualize the mathematical content in Section 2. In addition, whenever possible, we have omitted mathematical details and gave preference to illustrational figures.
>
> **The motivation of the paper is not clear.**
>
> In medical imaging, it is the boundary of a region which is annotated, and not the region itself. The raw ground-truth data is thus a discretization of a closed curve. The main motivation of our paper is to construct a model which (i) may directly use such raw ground-truth data if available (ii) is guaranteed to predict (smooth) curves (iii) improves inference speed by predicting 1d objects (curves) instead of 2d objects. We argue that in traditional pipelines, where pixel-based predictions are constructed, smooth boundaries are not faithfully represented. In particular, no prior information about the geometry of planar curves is incorporated.
>
> We have added this motivation to the introduction now (see the second paragraph).
>
> **Why is it subpixel?**
>
> Our model predicts a continuous representation of a curve, which may be evaluated at any point. Therefore, its range is not confined to an integer-lattice (pixels). This should be contrasted to a binary mask, which is confined to an integer-lattice.
>
> **The authors shall highlight their contributions and clarify the difference from the state-of-the-art algorithms.**
>
> The main contribution of our paper is the ability to parameterize boundaries of two-dimensional regions using continuous representations of curves (instead of discrete pixel-based approximations). This shift in point of view, to model a 1d object instead of 2d, allows for significant improvements in inference speed. Moreover, our models incorporate prior information about the geometry of planar curves, which typical  pixel-based models do not. This is accomplished by using multiresolution analysis and orthogonal series expansions of curves. While the technology of multiresolution analysis is well-established, to the best of our knowledge it has not been used yet to construct continuous representations of contours in a deep learning context.
>
> **It is not clear if the use of wavelets and multi-resolution analysis is a significant contribution or not. The improvement seems marginal compared with U-net**
>
> We believe that the incorporation of prior knowledge (the geometry of planar curves) to improve robustness, improvements in inference speed, and a sound theoretical foundation on which our decoder is built, together constitute a valuable contribution to the field of semantic image segmentation. In addition, it should be noted that the binary ground-truth matched by a traditional U-Net is a fundamentally different (easier) object than the continuous representation of a  curve matched by our network. Subtle curvature and geometry may be accurately presented using our ground-truth curves, e.g., by using a sufficiently large number of Fourier coefficients to compute approximation coefficients. Binary ground-truth masks, however, cannot capture such subtle geometry due to their discrete nature.
>
> **I suggest the authors to draw a diagram of their network.**
>
> We have added a schematic picture of the Pyramid Algorithm, i.e., the computational steps needed to go to a lower or higher resolution level, which forms the basis of our method. In addition, we have added a detailed picture of the architecture of our network, where the role of the Pyramid Algorithm as a decoder is explicitly depicted. In general, we have added figures to help the reader visualize the complete pipeline.
>
> **There are many methods for segmentation that shall be cited and compared.**
>
> We have added a more comprehensive list of papers to the bibliography. However, we stress that the focus of our paper is on a specific subset of segmentation methods, namely the prediction of contours. A comprehensive review of general segmentation methods is out of the scope of this paper.

---

### Official Review · Reviewer_ZJwW · 2021-11-02

**Correctness:** 3
**Technical Novelty And Significance:** 3
**Empirical Novelty And Significance:** 3
**Recommendation:** 6
**Confidence:** 5

**Main Review:**

Strengths:
1) In contrast to the related literature, where pixel-based segmentations are provided, the output of the proposed method is a closed curve.


Weaknesses
1) The extensive presentation of multi resolution analysis could be avoided as the theory is well known. More experiments could strengthen the paper.

2) A figure clarifying the network layers and the corresponding detail coefficients would be welcome to graphically explain the network architecture in section 3.2 and how the coefficients are associated with the skip connections.



**Summary Of The Paper:**

The paper proposes a method for boundary extraction from images using a U-net type network architecture. The boundaries are considered as closed curves represented by multiresolution analysis (wavelets). The down-sampling part of the network consists of a 2D encoder and the up-sampling part is a 1D decoder that reconstructs the boundaries.

**Summary Of The Review:**

Although an important part of the paper presents well-known theory, I like the idea of estimating closed curves and I suggest accepting the paper.

---

> ### Author Response · Authors · 2021-11-22
> **Thank you for your feedback!**
>
> Thank you for your feedback! We have given point-to-point answers to your feedback / questions below.
>
> **The extensive presentation of multi resolution analysis could be avoided as the theory is well known. More experiments could strengthen the paper.**
>
> We fully agree that the theory is well-known. However, in order to describe our network (the decoder in particular), the loss, associated hyperparameters, and additionally the result in Lemma A$3$ (used to compute approximation coefficients), we were unfortunately not able to avoid such a presentation. In addition, having a broad audience in mind, we decided to provide a more comprehensive background. Nonetheless, we have now reduced the review about MRAs to approximately 1 page (Sections 2.1 and 2.2). The subsequent material, spread out over the main body and appendix, is concerned with necessary modifications only, specific to our setting.
>
> **A figure clarifying the network layers and the corresponding detail coefficients would be welcome to graphically explain the network architecture in section 3.2 and how the coefficients are associated with the skip connections.**
>
> We have added a figure which depicts the architecture of our network in full detail. In particular, the role of the Pyramid Algorithm as a decoder, including the skip-connections and its relation to the detail coefficients, is explicitly depicted.

---

### Official Review · Reviewer_w2LA · 2021-11-03

**Correctness:** 4
**Technical Novelty And Significance:** 4
**Empirical Novelty And Significance:** 3
**Recommendation:** 6
**Confidence:** 3

**Main Review:**

Strengths:
1. It is the first work to construct pixel-independent representations of curves instead of pixel-based output, which provides new insight into using wavelet analysis on segmentation tasks.

2. The proposed method achieves up to 5x faster inference speed and competitive scores compared with a U-Net model.

Weaknesses:
1. The presentation of mathematic is complex and hard to read due to many notations. Also, it can be quite hard for readers to obtain a clear overview of the learning pipeline. It'd be better to demonstrate with figures.

2. It lacks the details of the decoder in the model, especially how the reconstruction process is performed in the decoder. It can be difficult to reimplement the model. It'd be better to demonstrate with figures.

3. The experiment results are not convincing enough. More experiments on other datasets are needed. The improvement of the proposed method is not obvious as compared with the baseline model. It lacks explanations on the reason for using these (j1,j2,j3) settings in the experiments.

4. It lacks explanations on the reason and motivation that it conducts wavelet analysis on the boundary instead of the object itself.

5. Some sentence is ambiguous.
In the first paragraph of Section 3, ‘…is henceforth assumed to have two components denoted by [y(x)]1 and [y(x)]2’ how is the two components generated?

6. Some typos are listed below.
In the abstract, ‘…the upsampling path is a one-dimensional decoder’, ‘a’ should be ‘an’.
In the second paragraph of subsection ‘detail coefficients’, ‘…states that the subspaces Wj can too be spanned by dilating and shifting a single map’, ‘too’ is redundant.
In the third paragraph of section 3.2, ‘The encoder is followed by a a bottleneck …’, ‘a’ is redundant.


**Summary Of The Paper:**

In this paper, the authors propose a novel deep learning framework for the fast prediction of boundaries. The boundary is modeled as smooth closed curves with wavelets and multi-resolution analysis. A U-shape model is trained to predict for coefficients of the curve. The proposed method is evaluated on medical images and achieves similar performances with up to 5x faster inference speed.
The main contribution is that it is the first to use wavelet analysis to construct pixel-independent representations of curves.


**Summary Of The Review:**

The paper provides novel insight into medical image segmentation. The technical introduction is sufficient and makes sense. The experimental results support the theoretical part.

It can be hard to read and reimplement. The improvement over the baseline is not obvious.

---

> ### Author Response · Authors · 2021-11-22
> **Thank you for your feedback!**
>
> Thank you for your feedback! We have used your comments to improve the presentation of our paper. Below we have summarized some point-to-point answers to your review:
>
> **Presentation mathematics complex and hard to read too much notation.**
>
> We have added schematic pictures to visualize the mathematical content in Section 2. In addition, whenever possible, we omit mathematical details and give preference to illustrative figures.
>
> **No clear overview of learning pipeline & Details decoder are missing (use figures)**
>
> We have added a schematic picture of the Pyramid Algorithm, i.e., the computational steps needed to go to a lower or higher resolution level, which forms the basis of our method. In addition, we have added a detailed picture of the architecture of our network, where the role of the Pyramid Algorithm as a decoder is explicitly depicted. In general, we have added figures to help the reader visualize the complete pipeline.
>
> **The experiment results are not convincing enough. More experiments on other datasets are needed** &
>   **The improvement of the proposed method is not obvious as compared with the baseline model**
>
> Unfortunately, due to time constraints, we have not been able to perform experiments on more datasets. However, we have added an additional experiment using a more advanced U-Net for comparison. In addition, we have made our code publicly available, see https://anonymous.4open.science/r/mra_segmentation/.
>
> **Paper lacks explanations on the reason and motivation that it conducts wavelet analysis on the boundary instead of the object itself**
>
> The reason is twofold. Firstly, in clinical practice (for instance in radiation oncology)  it is the boundary of a region that is annotated, not the region pixel-by-pixel. The annotation, and thus the raw ground-truth data, is interpreted as a discretization of a closed curve. Only after a contour has been delineated, the enclosed region is approximated and stored into an integer-lattice, which may not faithfully represent the raw ground-truth data anymore. In order to maintain the integrity of the ground-truth data, we argue that it is more natural to predict smooth curves directly from the start and if possible avoid any discretization steps in the ground-truth data.
>
> Secondly, the domains we consider, such as tumors or organs, are completely characterized by their boundaries. There is no need to explicitly parameterize the enclosed 2d region, which is computationally more expensive. In particular, many relevant questions regarding the enclosed region of interest, such as its area, midpoint, total amount of some quantity distributed over the region, etc., can be answered using a parameterization of the boundary, e.g., by integrating appropriate quantities over it (Green’s Theorem).
>
> We have incorporated the above motivation more clearly into the main body of the paper (see the introduction).
>
> **Some sentence is ambiguous. In the first paragraph of Section 3: it is henceforth assumed to have two components denoted by [y(x)]_1 and [y(x)]_2 how is the two components generated?**
>
> We tried to emphasize here that all analysis is henceforth performed on planar curves associated to (medical) images. The reason we stress this is at this point is because in the previous sections, we focused solely on the needed tools from wavelet theory as a subject in its own right. This exposition was set up in the context of a scalar-valued (periodic) signal (for notational convenience) and we did not refer to the context of predicting planar curves in 2d images until here in Section 3. We have now stated this distinction more clearly at the beginning of Section 2. Finally, it is explained in full detail in Section 4.2 how the ground-truth, i.e., planar curves with two components, are generated from a given discretization.
>
> **The upsampling path is a one-dimensional decoder’, ‘a’ should be ‘an’**
>
> In general, this is true for words starting with a vowel, but “one” is an exception apparently due to its pronunciation (consonant sound).
>
> **In the second paragraph of subsection ‘detail coefficients’, ‘…states that the subspaces $W_{j}$ can too be spanned by dilating and shifting a single map’, ‘too’ is redundant**
>
> It is not at all obvious that the subspaces $W_{j}$ may be spanned in the same way as the approximation subspaces $V_{j}$, i.e., by dilating and shifting a single map. We use the word “too” here to point out that the spaces $W_{j}$ may in fact also be spanned in such a manner.
>
> **In the third paragraph of Section 3.2, ‘The encoder is followed by a a bottleneck:  ‘a’ is redundant.**
>
> Section 3.2 has been thoroughly rewritten and is now explained via a schematic figure.
>
> **It can be hard to read and reimplement**
>
> We have made our code publicly available, see https://anonymous.4open.science/r/mra_segmentation/.

---

### Author Response · Authors · 2021-11-22
**Publicly available code**

We have made our code publicly available at https://anonymous.4open.science/r/mra_segmentation/

---

### Decision · Program_Chairs · 2022-01-20

**Decision:**

Reject

**Comment:**

The paper describes an interesting approach to predicting continuous closed surface segmentations from discretized image data using a wavelet output representation. This is an interesting idea with a lot of potential. Unfortunately, the paper currently suffers from major weaknesses which we encourage the authors to address.

1. While the idea of generating a continuous output representation of a segmentation is technically interesting, what are some applications where this is actually useful?
2. The ground truth annotations in the datasets evaluated are implicitly quite variable. The annotations are not made to sub-pixel precision and there is likely to be large multi-pixel variability across different annotations of the same image. This makes a poor problem to demonstrate the need and potential of a sub-pixel accurate segmentation algorithm.

I encourage the authors to find applications and datasets where reliable sub-pixel ground truth annotations exist, and to demonstrate that their approach to generating sub-pixel segmentations is superior to appropriate baselines which also predict sub-pixel segmentations.